# Repression of Irs2 by let-7 miRNAs is essential for homeostasis of the telencephalic neuroepithelium

Virginia Fernández[1,†] , Maria Ángeles Martínez-Martínez[1], Anna Prieto-Colomina[1],
Adrián Cárdenas[1], Rafael Soler[1], Martina Dori[2], Ugo Tomasello[1], Yuki Nomura[1], José P López-Atalaya[1],
Federico Calegari[2] & Víctor Borrell[1,*]

## Abstract

Structural integrity and cellular homeostasis of the embryonic stem cell niche are critical for normal tissue development. In the telencephalic neuroepithelium, this is controlled in part by cell adhesion molecules and regulators of progenitor cell lineage, but the specific orchestration of these processes remains unknown. Here, we studied the role of microRNAs in the embryonic telencephalon as key regulators of gene expression. By using the early recombiner *Rx-Cre* mouse, we identify novel and critical roles of miRNAs in early brain development, demonstrating they are essential to preserve the cellular homeostasis and structural integrity of the telencephalic neuroepithelium. We show that *Rx-Cre;Dicer^{F/F}* mouse embryos have a severe disruption of the telencephalic apical junction belt, followed by invagination of the ventricular surface and formation of hyperproliferative rosettes. Transcriptome analyses and functional experiments *in vivo* show that these defects result from upregulation of *Irs2* upon loss of *let-7* miRNAs in an apoptosis-independent manner. Our results reveal an unprecedented relevance of miRNAs in early forebrain development, with potential mechanistic implications in pediatric brain cancer.

**Keywords** apical adherens junction; cancer; ectopia; p53; Pax6
**Subject Categories** Neuroscience; RNA Biology
**The EMBO Journal (2020) 39: e105479**

## Introduction

The development of the telencephalon is a highly complex process involving a sequence of key events. Neurogenesis onsets with the emergence of apical radial glia cells (aRGCs), specialized neuroepithelial cells essential in telencephalic development that serve as neural progenitors and guide for the migration of newborn neurons. aRGCs are highly polarized and extend thin processes attached to the ventricular (apical) and pial (basal) surfaces of the developing telencephalon, with the cell bodies forming the ventricular zone (VZ). Their apical process terminates at the ventricular surface in an end foot, which serves to anchor aRGCs to each other via adherens junctions. This maintains the polarity of aRGCs and the cellular homeostasis and structural integrity of the VZ (Gotz & Huttner, 2005; Marthiens *et al*, 2010). Upon cell division, aRGCs generate either additional aRGCs, neurons, or basal progenitor cells, which delaminate from the ventricular surface and migrate to the basal side of the VZ, forming the SVZ (Miyata *et al*, 2004; Noctor *et al*, 2004, 2008). In mouse, most basal progenitors are intermediate progenitor cells (IPCs), producing the majority of excitatory neurons (Haubensak *et al*, 2004; Kowalczyk *et al*, 2009; Taverna *et al*, 2014).

The integrity of the VZ and its apical adherens junction belt is essential for the normal development of the telencephalon, including progenitor cell proliferation and neuron migration (Cappello *et al*, 2006; Rasin *et al*, 2007; Taverna *et al*, 2014). Its disruption due to developmental insult or genetic mutation leads to severe malformations of brain development in humans (Barkovich *et al*, 2012; Fernandez *et al*, 2016). Molecular mechanisms involved in the apical anchoring of aRGCs have been identified (Gotz & Huttner, 2005; Cappello *et al*, 2006; Rasin *et al*, 2007), but mechanisms regulating gene expression to preserve the global integrity and homeostasis of the neuroepithelial niche remain largely unexplored.

Gene expression and function during brain development are finely tuned by a number of regulatory mechanisms (Bae *et al*, 2015; Nord *et al*, 2015; Yao *et al*, 2016). Non-coding RNAs, and particularly microRNAs, are major post-transcriptional regulators of gene expression involved in many developmental processes (Aprea & Calegari, 2015). In the embryonic cerebral cortex, many cell cycle-related proteins are targets of miRNAs (Arcila *et al*, 2014). Previous mouse models of miRNA loss consistently used conditional mutants where miRNAs are depleted only at mid-late corticogenesis. This caused very limited defects on cell proliferation or neurogenesis, leading instead to massive apoptosis of progenitor cells and postmitotic neurons only at late developmental stages (De Pietri Tonelli

1  Instituto de Neurociencias, Consejo Superior de Investigaciones Científicas & Universidad Miguel Hernández, Sant Joan d'Alacant, Spain
2  CRTD-Center for Regenerative Therapies, School of Medicine, Technische Universität Dresden, Dresden, Germany
*Corresponding author. Tel: +34 965 919245; E-mail: vborrell@umh.es
†Present address: Fondazione Istituto Italiano di Tecnologia (IIT), Genoa, Italy

*et al*, 2008; Kawase-Koga *et al*, 2010; McLoughlin *et al*, 2012; Nigro *et al*, 2012; Saurat *et al*, 2013). As a result, miRNA function in early telencephalic development remains largely unknown.

Biogenesis of miRNAs requires processing of pre-miRNAs into mature miRNAs, which in turn depends on the action of the RNase enzyme Dicer1 (Bartel, 2018). Complete loss of *Dicer1* in full knockout mouse zygotes leads to early developmental defects and embryonic arrest after gastrulation, around embryonic day 7.5 (E7.5) (Bernstein *et al*, 2003), which evidences the fundamental importance of miRNAs in early embryonic development. Given this very early lethality, understanding the function of Dicer-dependent miRNAs in telencephalic development, which occurs much later, requires the use of conditional knockouts, crossed with a variety of Cre recombinase-expressing mouse lines (Harfe *et al*, 2005). *Emx1*-Cre, *Nestin*-Cre, and *hGFAP*-Cre mice have been widely used for studies of embryonic development of the cerebral cortex, taking advantage of their early expression (E9.5, E10.5, and E13.5, respectively) (Zimmerman *et al*, 1994; De Pietri Tonelli *et al*, 2008; Kawase-Koga *et al*, 2010; Saurat *et al*, 2013; Zhang *et al*, 2015). Elimination of Dicer with these Cre driver lines has produced a significant variety of phenotypes (Kawase-Koga *et al*, 2009), but surprisingly, those studies failed to identify significant roles for miRNAs in dorsal telencephalic development prior to E13.5, in spite of the high expression levels of miRNAs since E10.5 (Kloosterman *et al*, 2006; De Pietri Tonelli *et al*, 2008). This suggests that upon disruption of the gene locus, both Dicer protein and miRNA levels remain largely unchanged within targeted cells for a long time, or until diluted over consecutive cell cycles. Thus, elucidating the role of miRNAs in embryonic telencephalic development may require the removal of *Dicer* at much earlier stages than in previous studies.

Here, we have studied the role of miRNAs in early telencephalic development by using the *Rx-Cre* driver mouse line (Swindell *et al*, 2006), which recombines in the primordium of the telencephalon at E7.5, 3 days earlier than in previous models. *Rx-Cre;Dicer^{F/F}* (*Rx-Dicer*) mutant embryos displayed mild developmental defects in the neocortex related to increased apoptosis, similar to previous reports (De Pietri Tonelli *et al*, 2008). However, in the rostral telencephalon the early loss of *Dicer* led to very severe tissue disorganization and the massive formation of highly proliferative rosettes, which grew caudally. Time-course analyses revealed that rosettes formed by invagination of the rostral neuroepithelium, a process concomitant with apoptosis but independent from it, and due to the loss of apical adherens junctions and increased proliferation. These two aspects of the phenotype emerged from decreased levels of *let-7* miRNAs and increased expression of targets that promote apoptosis and/or proliferation: *p53* signaling and i*nsulin receptor substrate-2* (*Irs2*), respectively. The formation of rosettes in *Rx-Dicer* mutants was prevented by the loss of p53, but this was independent from a loss in apoptosis, as overexpression of *Irs2* alone in wild-type embryos was sufficient to induce rosette formation without massively increasing apoptosis. The formation of hyperproliferative rosettes upon *Irs2* overexpression was rescued by overexpression of *let-7*. This was phenocopied by the loss of endogenous *let-7* alone, which was then rescued by the loss of function of Irs2. The positive effects of Irs2 on telencephalic progenitor proliferation, and negative of *let-7*, were phenocopied in human cerebral organoids, indicating that this is a highly conserved mechanism. Our results suggest a general

relevance of miRNA dysregulation on the emergence of malformations of early brain development and, potentially, other tissues of ectodermal origin.

# Results

### Early loss of telencephalic miRNAs in *Rx-Cre;Dicer^{F/F}* embryos

To investigate the roles of miRNAs in early telencephalic development, we circumvented the timing limitation of previous Cre driver lines by using *Rx-Cre* mice, which express Cre under the control of the regulatory sequences of the transcription factor *Rax* (*Rx*) (Swindell *et al*, 2006). Best known for its specific expression in the developing retina, *Rx* is first expressed in the anterior neural fold (prospective forebrain) of E7.5 mouse embryos, 3 days earlier than *Emx1* (Furukawa *et al*, 1997). *Rx-Cre* mice crossed with the *Rosa26-tdTomato* reporter line (Madisen *et al*, 2010) demonstrated Cre recombination in the emerging telencephalic vesicles as early as E8.5, later becoming distinctively restricted to the telencephalon (Fig EV1A). Within the telencephalon, tdTomato expression level changed as development progressed, gradually increasing from E11.5 to E17.5/postnatal day 1 (P1; Fig EV1B–D). The basal ganglia expressed the highest levels of tdTomato already at E11.5, with the olfactory bulb (OB) and septum reaching similar levels by E12.5 (Fig EV1C and D). In contrast, tdTomato expression at these early stages was very low in the neocortex, indicating a significantly lower level of Cre recombination (Fig EV1C and D, Table EV1). TdTomato levels in the neocortex increased gradually from E12.5 to E17.5, when reaching statistical similarity with the rest of the telencephalon (Fig EV1D). Nevertheless, the levels of tdTomato expression in the neocortex always tended to be lowest in the caudal and highest in the rostral neocortex, the area where expression reached greater similarity with the OB, septum, and basal ganglia. TdTomato in the thalamus was virtually absent at all ages, in agreement with this region not deriving from the lineage of $Rx^+$ territories (Fig EV1D). Taken together, these analyses showed that Cre recombination in Rx-Cre mouse embryos is most prevalent in the rostral and ventral telencephalon at E11.5 and E12.5, while it is significantly lower in the neocortex, particularly in its medial and caudal aspects.

Next, we tested the elimination of Dicer-dependent miRNAs using a conditional *Dicer^{flox/flox}* mouse line (*Dicer^{F/F}*). ISH against *Dicer* and *miR9*, one of the miRNAs most highly expressed in the developing vertebrate brain (Radhakrishnan & Alwin Prem Anand, 2016), demonstrated their substantial reduction in the rostral telencephalon of *Rx-Cre;Dicer^{F/F}* mutants at E12.5, and their absence by E17.5 (Fig EV2). At early developmental stages, the loss of *Dicer* and *miR9* was almost complete and much more prominent in the rostral and ventral telencephalon (OB, septum, and basal ganglia) than in the dorsal telencephalon (neocortex), consistent with the pattern of TdTomato expression upon Rx-Cre recombination described above. *Dicer* and *miR9* were still expressed at medial levels of the dorsal telencephalon by E17.5 (Fig EV2), indicating an incomplete recombination of *Dicer* in this territory. Given the high abundance of *miR9* in the normal mouse embryo brain, its major reduction in expression confirmed the loss of functional Dicer in *Rx-Cre;Dicer^{F/F}* embryos.

## Massive and transient cell death without loss of proliferation in the rostral telencephalic primordium

Previous *in vivo* and *in vitro* studies demonstrated that *Dicer* mutants typically exhibit high levels of cell death, including in the developing neocortex, particularly at intermediate to late embryonic stages (Mott *et al*, 2007; Raver-Shapira *et al*, 2007; Davis *et al*, 2008; De Pietri Tonelli *et al*, 2008). In contrast, our analysis of Casp3 stains in the E11.5 telencephalic primordium of *Rx-Cre;Dicer^{F/F}* mutant embryos (*Rx-Dicer* mutants, from hereon) revealed the occurrence of dramatically high levels of apoptosis in the rostral and ventral telencephalon, but not in the neocortex (Fig 1A). This was consistent with the previous reports showing that miRNAs are critical to prevent apoptosis in the developing telencephalon, but in this case at much earlier stages, and also consistent with the greater level of Cre recombination in the rostral and ventral embryonic telencephalon of *Rx-Cre* mice (Figs EV1 and EV2). Marker analyses showed that apoptosis involved mostly Pax6$^+$ aRGCs (67% of Casp$^+$ cells) and only a small minority of Tbr1$^+$ neurons (8.6% of Casp$^+$ cells; Fig 1B and C). A detailed time-course analysis revealed that apoptotic events started suddenly and at high levels at E11.5, with apoptotic cells arranged in columns that spun the entire thickness of the telencephalic primordium (Fig 1D and E). Remarkably, massive cell death lasted only between E11.5 and E12.5, decreasing suddenly again by E13.5 (Fig 1D and E).

Next, we reasoned that the dramatically high levels of apoptosis among aRGCs would strongly decrease proliferation. However, anti-PH3 stains demonstrated that the density of apical and basal mitoses was not significantly altered in *Rx-Dicer* mutant embryos between E10.5 and E13.5 (Fig 1F and G). Together, our results indicated that *Rx-Dicer* mutant embryos are severely affected by massive apoptosis largely in the rostral telencephalon and between E11.5 and E12.5, affecting mostly aRGCs. Remarkably, in spite of this massive cell death, the density of mitotic events remained unaltered in *Rx-Dicer* mutants, suggesting that the remaining non-apoptotic progenitor cells may proliferate and self-renew at rates higher than normal.

## Disorganization of the rostro-ventral telencephalon in *Rx-Dicer* embryos

Next, we investigated the long-term consequences of the high apoptotic levels in the rostral telencephalic primordium (prospective OB) of early *Rx-Dicer* mutant embryos. Sagittal sections through the brains of *Rx-Dicer* mutants at E17.5 showed that the OB was much smaller than in control embryos (Fig 2A–C). We confirmed the OB identity and reduction in size by ISH stains for *Grm1*, a marker of mitral cells in E17.5 embryos, which also revealed that the laminar organization of the OB was largely preserved (Fig 2B and C). In addition to a smaller OB, we observed a general and profound disorganization of the entire rostral–ventral region of the telencephalon in *Rx-Dicer* mutants, including the prefrontal neocortex and the septum (Fig 2D). These alterations also affected the rostral neocortex, but not its parietal region (Fig EV3A and B), consistent with the greater loss of miRNAs at early embryonic stages in the former. The disorganization of the rostral telencephalon in *Rx-Dicer* mutants involved an overabundance of Ki67$^+$ progenitor cells in the germinal zones (Fig 2D). A closer examination revealed that Ki67$^+$ cells were in fact exquisitely arranged in proliferative rosettes (Fig 2E

and F). This was striking because proliferative rosettes have never been reported in any of the previous *Dicer* mutant mouse lines (Kloosterman *et al*, 2006; De Pietri Tonelli *et al*, 2008).

Rosettes had a distinct laminar organization with apical–basal polarity, reminiscent of the telencephalic germinal layers in normal development. An inner lumen was delimited by a pseudostratified layer of progenitor cells undergoing apical mitosis, this was surrounded by a band of progenitor cells undergoing basal mitoses, and finally, these were enclosed by Tuj1$^+$ neurons (Fig 2E). Brain clarification experiments showed numerous rosettes located in the rostral and ventral part of the telencephalon, most with a spherical shape but some tubular, following either horizontal or vertical orientations (Fig 2F). Progenitor cells within the rosettes of E17.5 *Rx-Dicer* mutants displayed 10% increase in BrdU incorporation and 95% increase in cell cycle re-entry compared with the rostral cortex of age-matched control littermates (Fig 2G and H), indicating hyperproliferative activity. Rosettes were evident already at E14.5, when their apical surface displayed many features typical of the telencephalic ventricle, such as lining with apical complex proteins (Par3), adherens junction proteins (β-catenin), Arl13b$^+$ primary cilia and Phospho-Vimentin$^+$ apical progenitor cell bodies (Fig 2I) (Gotz & Huttner, 2005). These are typical features of true ependymal rosettes found in human embryonic tumors (Korshunov *et al*, 2010). These results demonstrated that functional Dicer is important in the murine early embryonic telencephalon to limit the rate of self-amplification of neural progenitor cells and to ensure the structural and cellular homeostasis of the telencephalic neuroepithelium.

The rostral–ventral telencephalon of *Rx-Dicer* mutants was severely disorganized already at E14.5 due to the high abundance of rosettes (Fig 2J). To investigate their regional identity, we analyzed the expression of specific marker genes (Fig 2K). *Pax6*, *Tbr2*, and *Ngn2* are transcription factors expressed at high levels by progenitor cells in the pallium (dorsal telencephalon) but not the subpallium (ventral telencephalon) (Englund *et al*, 2005; Flames *et al*, 2007; Diez-Roux *et al*, 2011). We found that rosettes located in pallial regions expressed these three pallial marker genes, while those located in subpallial regions did not (Fig 2K–N and P). Conversely, rosettes in ventral regions expressed *Dlx2* and *Dlx5*, two pan-ventral markers expressed in VZ + SVZ (*Dlx2*) or only in SVZ (*Dlx5*; Fig 2K and Q) (Diez-Roux *et al*, 2011). A detailed analysis of ventral rosettes revealed that those in rostral parts expressed *Gsx2* and low levels of *Pax6* (Fig 2M and O), typical of the developing subpallial septum (Flames *et al*, 2007). Rosettes found in caudal parts reached well into the basal ganglia: In the MGE, they were positive for *Nkx2.1* (Fig 2R), a marker of this region (Diez-Roux *et al*, 2011). In the VZ of LGE, rosettes did not express *Gsx2* (marker of this region) but maintained a subpallial identity (*Dlx2/5$^+$*; Fig 2K, Q and S) (Waclaw *et al*, 2009; Diez-Roux *et al*, 2011). Taken together, these results showed that *Rx-Dicer* mutant embryos develop hyperproliferative rosettes in multiple territories of the rostral and ventral telencephalon that maintain their territorial genetic identity.

## Rosettes mature and grow upon invagination from the VZ

Next, we sought to understand the histogenetic mechanisms leading to the formation of rosettes in *Rx-Dicer* mutants. A careful examination led us to distinguish three main types of rosette morphology (Fig 3A and B): Type 1 rosettes were a local invagination of the VZ

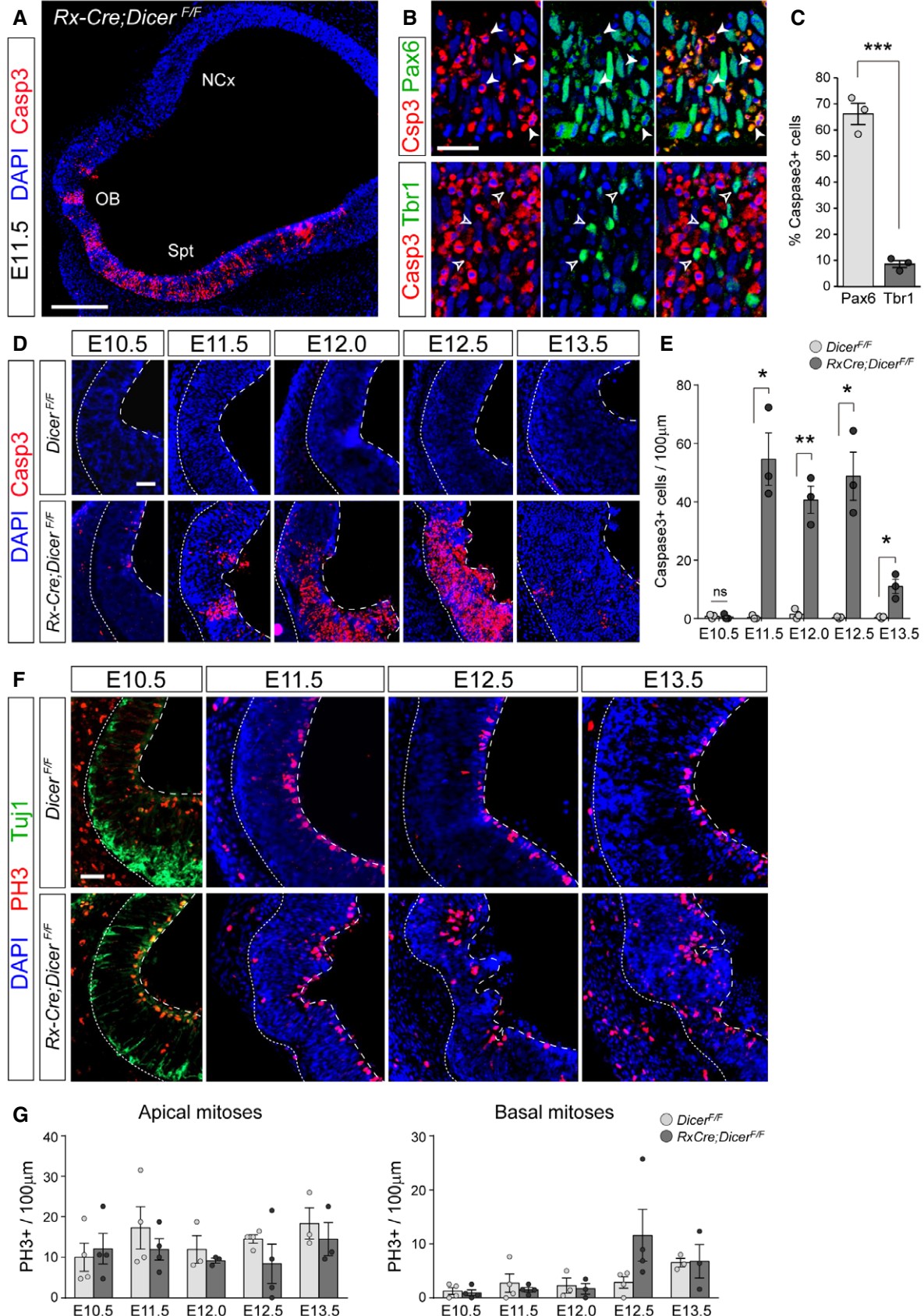

Figure 1.

◄

**Figure 1.    Massive progenitor cell apoptosis without global loss of proliferation in rostral telencephalon of *Rx-Dicer* mutants.**

A        Distribution of Casp3$^+$ cells in rostral and ventral domains of an E11.5 *Rx-Dicer* mutant embryo. NCx, neocortex; OB, olfactory bulb; Spt, septum.

B, C    Marker analysis of Casp3$^+$ cells in the rostral telencephalon of E12.5 *Rx-Dicer* mutant embryos. Most apoptotic cells are Pax6$^+$ RGCs (solid arrowheads) and not Tbr1$^+$ neurons (open arrowheads). N = 3 replicates per marker.

D, E    Distribution and abundance of apoptotic cells (Casp3$^+$) in the rostral telencephalic primordium of control and *Rx-Dicer* mutant embryos at the indicated ages. Dotted line indicates basal surface, and dashed line, apical surface. N = 3 replicates per genotype and age.

F, G    Distribution and abundance of mitoses (PH3$^+$) and neurons (Tuj1) in the rostral telencephalic primordium of control and *Rx-Dicer* mutant embryos at the indicated ages. Dotted line indicates basal surface, and dashed line, apical surface. N = 3–4 replicates per genotype and age. No significant differences were found between control and mutant embryos in apical nor basal mitoses.

Data information: Histograms represent mean ± SEM; symbols in plots indicate values for individual embryos; chi-square test (C), *t*-test (E, G); *$P < 0.05$, **$P < 0.01$, ***$P < 0.001$.

Scale bars, 200 μm (A), 40 μm (B), 25 μm (D, F).

apical side, frequently next to Tuj1$^+$ neurons that penetrated the VZ from the basal side to the apical surface. Type 2 rosettes were characterized by a profound invagination of the VZ with an inner lumen isolated from the telencephalic ventricle, but with the apical surface of both sides in continuum, identified by a line of apical PH3$^+$ nuclei (Fig 3A and B). These rosettes were surrounded by a layer of Tuj1$^+$ neurons and a prominent band of basal mitoses in between. Type 3 rosettes shared the same characteristics as type 2 except that the inner lumen was completely disconnected from the telencephalic ventricle, and they were always at a distance from it (Fig 3A and B). The relative frequency of rosette types varied with developmental age (Fig 3C). Type 1 was most abundant at E12 (70%), which gradually decreased to 30% at E14.5, and was absent at later stages. Conversely, type 3 rosettes were absent in E12.5 embryos and their frequency increased at later stages, to represent 95% of rosettes at E17.5 (Fig 3C).

Next, we studied the distribution of rosettes along the rostral-to-caudal axis of the developing telencephalon of *Rx-Dicer* mutant embryos (Fig 3D). At E12.5, when most rosettes are type 1, the majority (75%) were located at the most rostral level of the telencephalon and none at intermediate or caudal levels. At E13.5 and E14.5, rosettes were found along the rostral and intermediate regions with similar abundance, but none was observed at caudal levels. In the caudal-most area of the telencephalon, we only found rosettes at late stages, particularly abundant at E17.5 (Fig 3D). Since at E17.5 we did not observe type 1 rosettes, those found at caudal levels may have originated in rostral or intermediate levels and grow caudally. Considering together the temporal and regional variation in rosette types and their abundance, a parsimonious interpretation of our analyses is that those represent distinct phases of rosette maturation. Starting as small invaginations of the VZ (type 1), the high proliferative activity of Dicer mutant progenitors may cause a progressive growth and ingression (type 2), to finally detach from the telencephalic ventricle and become an isolated structure inside the brain parenchyma (type 3), where its growth continues toward the caudal telencephalon (Fig 3E).

## Loss of *Dicer* leads to disruption of apical junctions and formation of neuronal ectopias

The above results showed that *Dicer* ablation by *Rx-Cre* recombination resulted in a severe reduction in OB size and formation of rosettes in the rostral–ventral telencephalon circa E12.5, from where they extended to caudal regions. We next investigated the cellular mechanisms leading to the initial formation of rosettes, focusing on

the rostral-most aspect of the telencephalon between E10.5 and E13.5. Given that rosette formation involved the invagination of the VZ frequently flanked by ectopic neurons, we first focused on those neurons. Between E11.5 and E12.5, the rostral telencephalon of *Rx-Dicer* mutant embryos exhibited clusters of ectopic Tuj1$^+$ neurons accumulating within the VZ, away from their normal basal location and especially abundant at the apical side, even inside the ventricular lumen traversing the apical surface (Fig 3F and G). The apical surface of the embryonic telencephalon is lined with a belt of apical adherens junction (AJ) proteins, including Par3, β-catenin, and others, which anchor aRGCs together within the VZ (Gotz & Huttner, 2005). This AJ belt maintains the structural integrity and apical–basal organization of the VZ. The presence of neuronal ectopias penetrating this ventricular lining, and even invading the ventricular lumen, indicated that the AJ belt might be compromised in *Rx-Dicer* mutants. Analysis of Par3 protein distribution confirmed the severe disruption of the AJ belt in mutant embryos starting at E12.0 (coincident with the onset of rosettes), where this ventricular lining was frequently interrupted and formed ectopic clusters within the cortical parenchyma (Fig 3G). Apical zones where Par3 was discontinuous or absent were systematically populated by ectopic neurons. Overall, the rostral neuroepithelium of *Rx-Dicer* mutants was not only highly disorganized but much thinner than in control embryos, with a wrinkled ventricular surface (Fig 3F and G).

## Defective expression of *let-7* miRNAs and p53 pathway genes in early *Rx-Dicer* mutants

To unravel the genetic origin of rosettes developing in the rostral telencephalon of *Rx-Dicer* mutant embryos, we compared the transcriptome of mutants and control littermates. We performed this analysis at E11.5, the time when defects in AJs and apoptosis were first detected, and immediately prior to the emergence of rosettes. RNA-Seq analysis of miRNAs revealed significant reductions in expression levels for only 11 mature miRNAs (Fig 4A, Table EV2). The majority of downregulated miRNAs (eight out of 11) were members of the *let-7* family, expressed at two- to fourfold lower levels in E11.5 *Rx-Dicer* mutants compared with control littermates (Fig 4A and B, Table EV2). Using seed sequence-based analyses, we identified the mRNAs predicted to be direct targets of the miRNAs differentially expressed in *Rx-Dicer* mutants (Table EV3). Gene Ontology analysis of the predicted target gene set revealed an enrichment in terms related to regulation of transcription, response to DNA damage and apoptosis, positive regulation of cell proliferation, cell cycle re-entry, and cell–cell adhesion (Fig 4C, Table EV4).

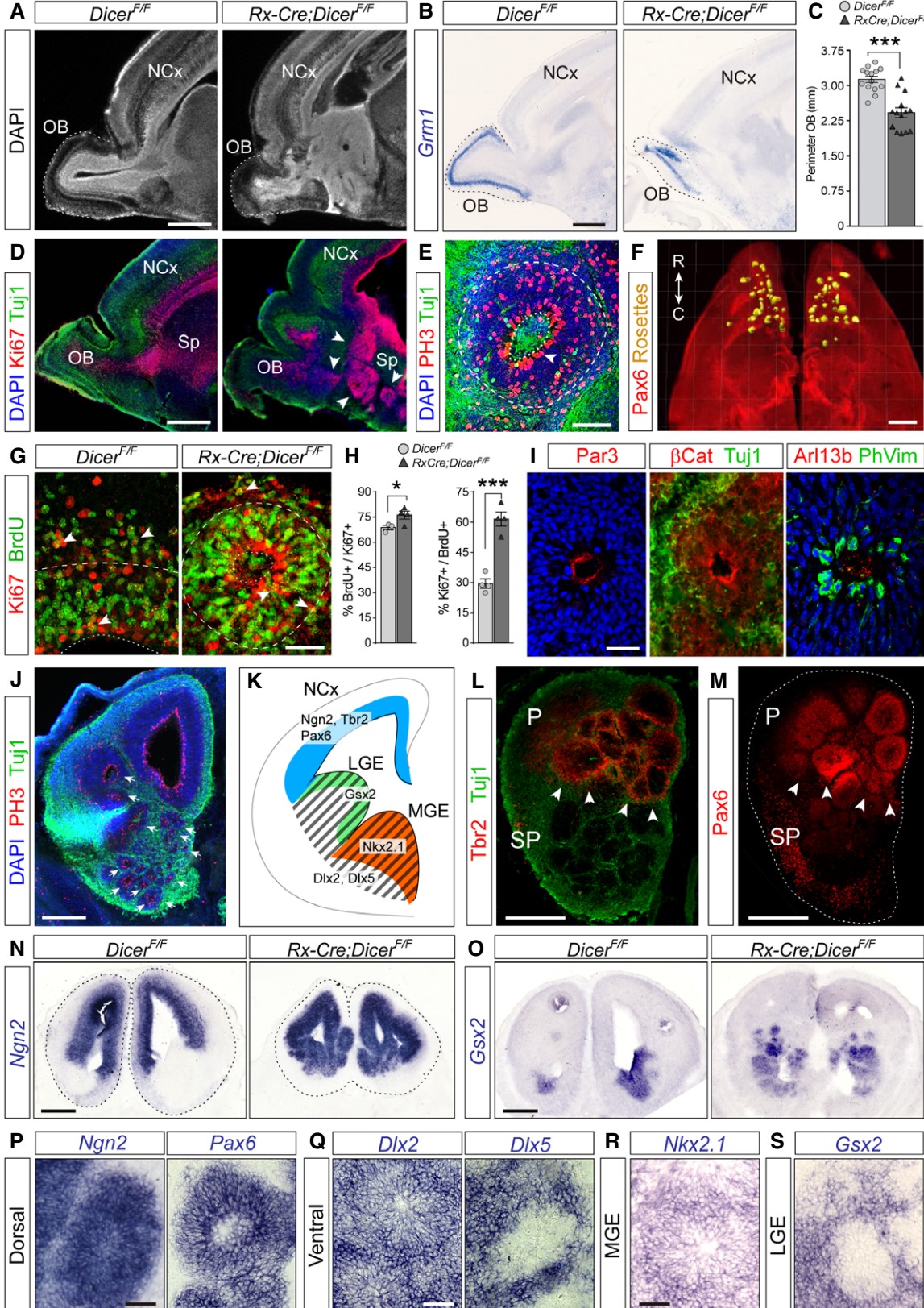

**Figure 2.**

**Figure 2. Formation of rosettes in the rostral telencephalon of *Rx-Dicer* mutants.**

A   DAPI stain of sagittal sections through the rostral telencephalon of E17.5 control and *Rx-Dicer* mutant littermates showing the neocortex (NCx) and olfactory bulb (OB; dashed line).

B   Expression pattern of *Grm1* mRNA in OB (dotted line) of control and *Rx-Dicer* mutants.

C   Quantification of OB perimeter (mean ± SEM; symbols indicate values for individual embryos); *t*-test, ***$P < 0.001$. $N = 14$ replicates per genotype.

D   Immunostains of E17.5 control and *Rx-Dicer* mutant brains showing the distribution of progenitor cells (Ki67, red) and neurons (Tuj1, green). Arrowheads indicate rosettes. Sp, septum.

E   Detail of a rosette from an E17.5 *Rx-Dicer* mutant embryo displaying typical features: closed apical surface (dotted line) with PH3+ apical mitoses (white arrowhead) and basal mitoses, surrounded by Tuj1+ neurons (green). Dashed line indicates the basal border of the rosette.

F   Rostral half of an *Rx-Dicer* mutant E17.5 brain immunostained for Pax6, clarified, and segmented to reveal rosettes (yellow). C, caudal; R, rostral.

G, H   BrdU incorporation and cell cycle re-entry analysis with the progenitor cell marker Ki67 at E17.5, in the rostral cortex of control embryos compared with rostral rosettes of *Rx-Dicer* mutants. Dotted lines indicate apical surface, and dashed lines indicate basal border of VZ. Arrowheads indicate double-positive cells. Data in histograms are mean ± SEM, symbols indicate values for individual embryos; $n \geq 3$ embryos; chi-square test, *$P < 0.05$, ***$P < 0.001$.

I   Apical lumen of rosettes immunostained against apical complex proteins (Par3, β-catenin), primary cilia (Arl13b), apical mitoses (PhVim), and neurons (Tuj1).

J   Coronal section through the rostral telencephalon of an E14.5 *Rx-Dicer* mutant embryo illustrating the high abundance and location of rosettes (arrows), as revealed by the distribution of mitoses (PH3) and neurons (Tuj1).

K–S   Analysis of the regional identity of rosettes in E14.5 *Rx-Dicer* mutants. (K) Schema of normal transcription factor expression patterns defining telencephalic regional identity. Expression of *Ngn2*, *Tbr2*, and *Pax6* identifies rosettes in the rostro-dorsal telencephalon as having dorsal identity (L–N, P); *Gsx2*, *Dlx2*, and *Dlx5* identify rosettes in the ventral telencephalon as having ventral identity (O, Q); *Nkx2.1* identifies MGE rosettes as being normotopic (R); the absence of *Gsx2* in LGE rosette cells identifies them as ectopic (S). Tuj1 labels neurons. In (L, M), arrowheads indicate the border between dorsal and ventral territories, and dotted line indicates the outer border of the telencephalon. LGE, lateral ganglionic eminence; MGE, medial ganglionic eminence; P, pallium; SP, subpallium.

Data information: Scale bars, 500 μm (A, B, D, F), 25 μm (E, G, I, P–S), 200 μm (J–O).

These functional terms were independently confirmed by the analysis of signaling pathways, which highlighted that our set of target genes are involved in several cancer pathways including glioma, pathways promoting stem cell pluripotency and proliferation, focal adhesion, and p53 signaling (Fig 4C, Table EV5). Salient functional terms and signaling pathways were consistent with the rosette phenotype we observed in *Rx-Dicer* mutants, pointing to a direct role of the differentially expressed miRNAs (mostly *let-7*) in the maintenance of the homeostasis of the telencephalic neuroepithelium, and thus the emergence of rosettes in their absence.

RNA-Seq analysis of protein-coding genes revealed 542 transcripts differentially expressed between mutant and control littermates (DEGs; *P*adj < 0.05, fold change > ± 1.25; Fig 4D, Table EV6). As expected by a loss in miRNAs as negative regulators of mRNA abundance, a majority of DEGs were upregulated (59%) and their average fold change (log$_2$FC = 1.026 ± 0.046) was greater than among downregulated genes (41%, log$_2$FC = −0.685 ± 0.042; $P = 4,5e$-103, *t*-test). Among the top 5% of DEGs most significantly changed, we found *Prtg*, *Lin28*, *Greb1*, and *Irs2* (Fig 4E), known as positive regulators of cell proliferation upregulated in cancer (Wong *et al*, 2010; Stamateris *et al*, 2016; Farzaneh *et al*, 2017; Hodgkinson *et al*, 2018). Gene Ontology analyses of DEGs revealed that the terms most highly enriched were related to regulation of transcription, cell differentiation, regulation of proliferation, cell adhesion, cell migration, and epithelial cell development (Fig 4F, Table EV7). In agreement with these GO terms, KEGG pathway analysis highlighted pathways involved in cancer and progenitor cell proliferation (Shh, PI3K-Akt, ECM), as well as in apoptosis (p53; Fig 4F, Table EV8). Accordingly, miRNAs of the *let-7* family are downregulated in various types of cancer (Johnson *et al*, 2005; Roush & Slack, 2008) and several regulate p53 signaling, cell proliferation, and/or tumorigenicity (Yu *et al*, 2007; Suh *et al*, 2012; Wang *et al*, 2012; Li *et al*, 2014; Subramanian *et al*, 2015). These results revealed a transcriptomic landscape in E11.5 *Rx-Dicer* mutant embryos consistent with our observed phenotypes of abundant cell death and formation of rosettes with high cell cycle re-entry. *Ad hoc* analyses of differential expression confirmed the occurrence of transcriptomic changes

directly or indirectly involved in regulation of cellular apoptosis and proliferation (Fig 4G). Regarding increased apoptosis, this included augmented expression of *Rassf3*, *Trp53inp1*, *Sesn2*, and *Zmat3* and reduced levels of *Prok2* and *Atf5* (Kudo *et al*, 2012; Juliana *et al*, 2017; Zhang *et al*, 2019), whereas increased levels of *Ptch1*, *Shh*, *Cdk6*, *Pten*, and *Irs2* and reduced levels of *Insm1* and *Pax6* were consistent with augmented apical progenitor cell amplification (Gritli-Linde *et al*, 2002; Farkas *et al*, 2008; Wong *et al*, 2015; Stamateris *et al*, 2016; Tavano *et al*, 2018). Gene set enrichment analysis confirmed the genetic upregulation of the same and related processes and signaling pathways (Fig 4H and Tables EV9 and EV10), further supporting their involvement in apoptosis and formation of proliferative rosettes in *Rx-Dicer* mutants. Furthermore, a number of both differentially expressed mRNAs and target genes of differentially expressed miRNAs were functionally related to cell adhesion, thus possibly contributing to alter the apical adherens junction belt and the formation of rosettes. Functional annotation of targets of differentially expressed miRNAs revealed many GO categories related to adhesion including "Heterophilic cell-cell adhesion via plasma membrane cell adhesion molecules" and "Substrate adhesion-dependent cell spreading", and significant enrichment for the KEGG pathway "Focal adhesion" (Table EV11). Similarly, many differentially expressed mRNAs in Rx-Dicer mutant embryos were functionally related to the GO terms "Cell adhesion", "Cell–cell adhesion", "Single organismal cell-cell adhesion", "Heterophilic cell–cell adhesion via plasma membrane cell adhesion molecules", and "Extracellular matrix-receptor interaction". GSEA of DEGs was consistent with the above results, revealing significant enrichment for the GO categories "Biological adhesion", "Positive regulation of cell adhesion", "Regulation of cell adhesion", and "Cell–cell adhesion", whereas GSEA for KEGG pathways identified genes associated with "Focal adhesion" and "Adherens junction" (Table EV11). Upregulation of *Irs2* and downregulation of *let-7* expression in the rostral telencephalon of *Rx-Dicer* mutant embryos compared with controls was confirmed by *in situ* hybridization and/or quantitative PCR (*Irs2*, 2.35-fold increase, $P < 0.01$; *let-7*, 2.34-fold decrease; Fig 4I and J).

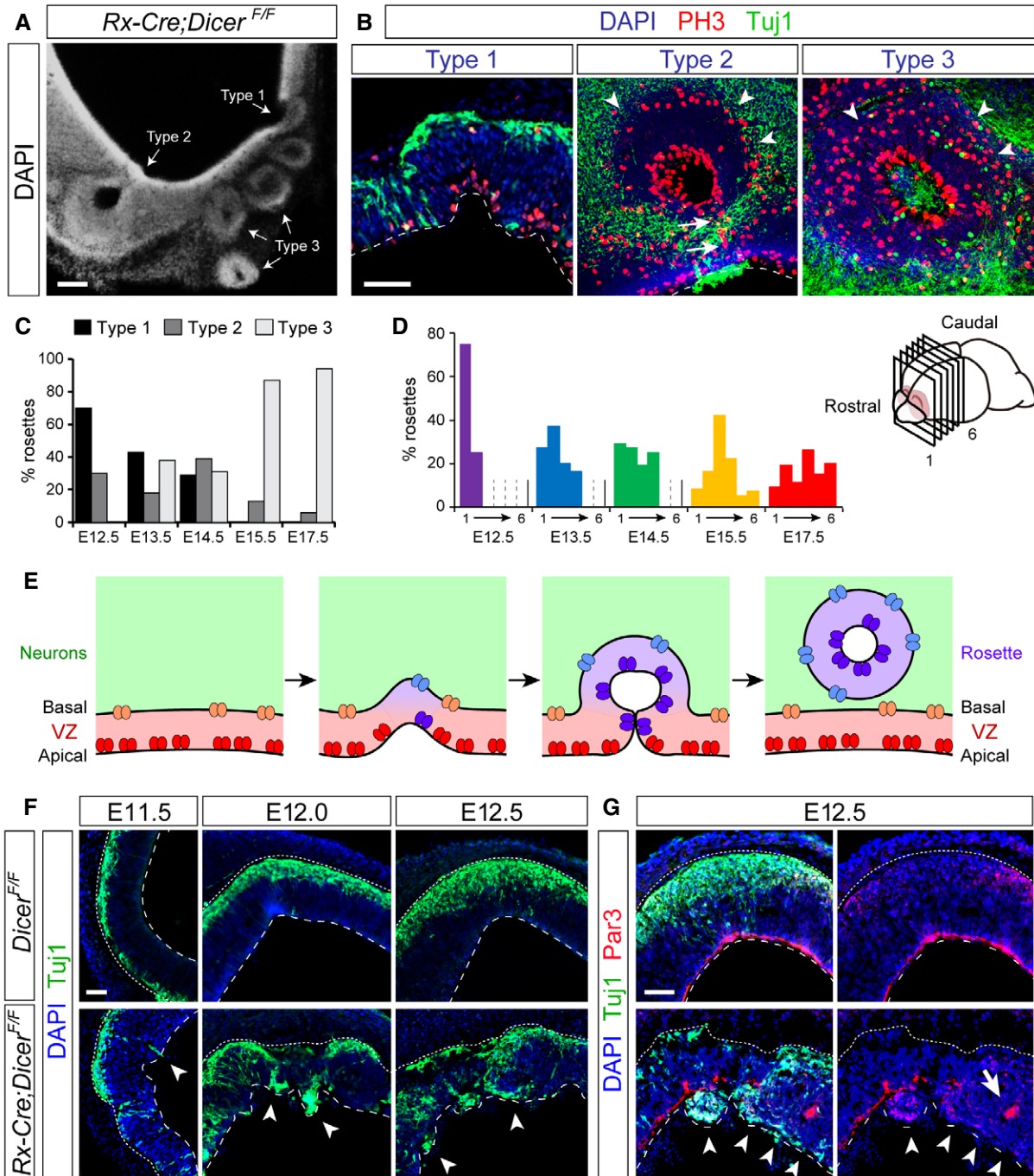

**Figure 3.   Disturbance of ventricular integrity precedes the formation of rosettes.**

A       Coronal section through the rostral telencephalon of an E14.5 *Rx-Dicer* mutant embryo displaying the three types of rosettes.

B       High magnification of rosettes in *Rx-Dicer* mutants at E12.5 (type 1), E14.5 (type 2), and E17.5 (type 3), immunostained for PH3 and Tuj1. Dashed line indicates ventricular surface; arrowheads indicate basal mitoses; arrows indicate a stream of apical mitoses connecting the lumen of the rosette with the lumen of the telencephalic ventricle.

C, D    Abundance of rosette types at the indicated ages (C), and rostro-caudal distribution of total rosette abundance per age, independent of type (D). *n* ≥ 2 brains per age.

E       Schematic of the progression of rosette formation in the rostral telencephalon of *Rx-Dicer* mutant embryos.

F, G    Distribution of neurons (Tuj1⁺ cells) and apical adherens junction protein Par3 in the OB primordium of control and *Rx-Dicer* mutant embryos at the indicated ages. Dotted lines indicate basal border, and dashed lines indicate apical border. White arrowheads indicate ectopic neurons and/or the absence of Par3. Arrow in (G) indicates accumulation of Par3 at the lumen of a nascent rosette.

Data information: Scale bars, 100 μm (A), 40 μm (B), 50 μm (F, G).

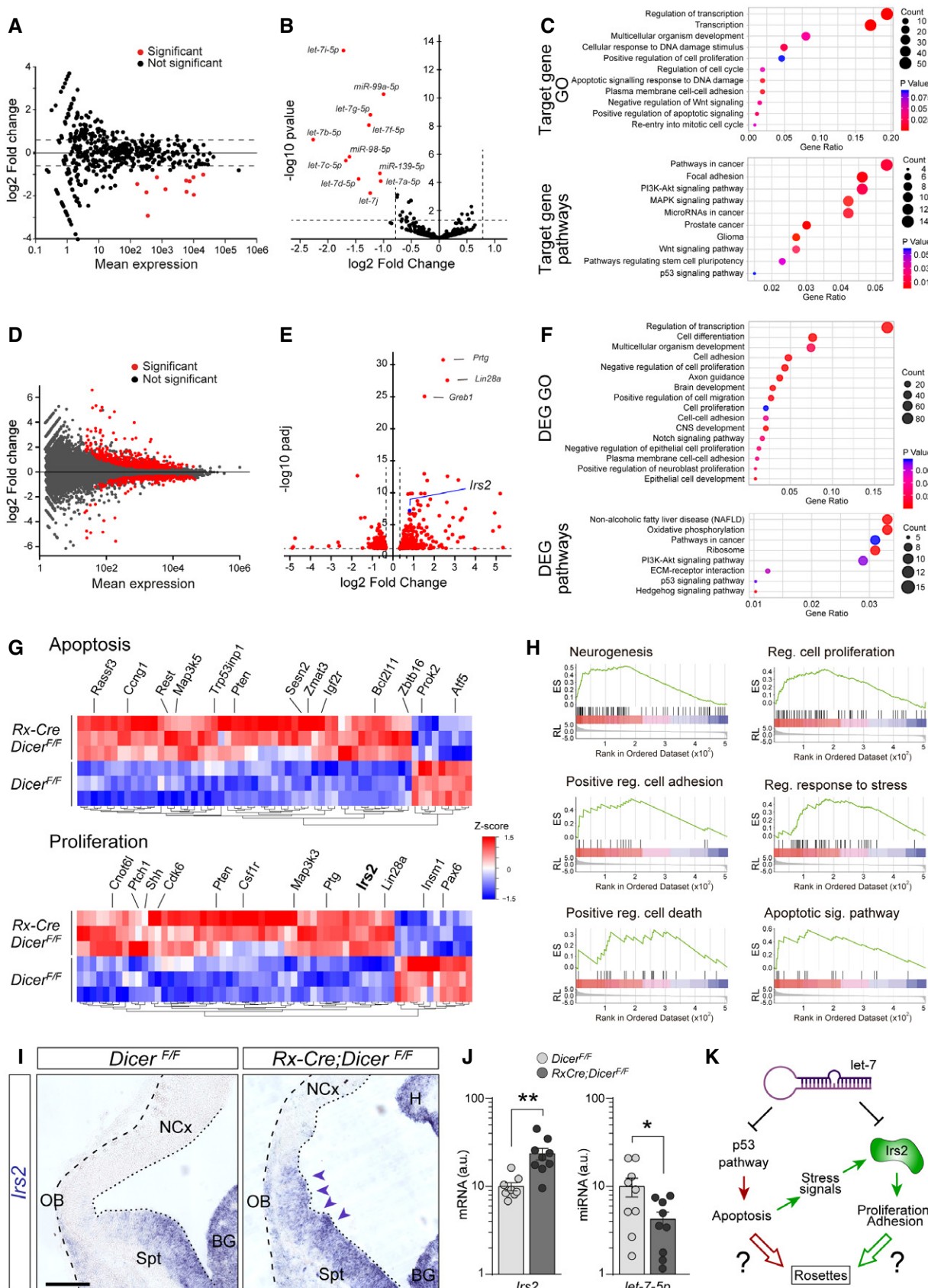

Figure 4.

**Figure 4. Changes in expression of miRNAs and mRNAs upon early loss of Dicer.**

A, B   Changes in expression levels of miRNAs between control and *Rx-Dicer* mutants at E11.5 in the rostral telencephalon. Red dots indicate statistically differentially expressed miRNAs according to FDR correction of *P* values based on the Wald statistic (DESeq2 analysis) (B). Dashes lines indicate stringency limits in evaluation of results.

C   Gene Ontology terms (top) and signaling pathways (bottom) associated with target genes of the miRNAs differentially expressed between control and *Rx-Dicer* mutants.

D, E   Changes in mRNA expression levels between control and *Rx-Dicer* mutants at E11.5 in the rostral telencephalon. Statistically differentially expressed mRNAs according to FDR correction of *P* values based on the Wald statistic (DESeq2 analysis; Adj. *P* < 0.1 and FC > 1.25) are highlighted in red, and the three with the highest Adj. *P* value are identified by name. Blue dot indicates *Irs2*: FC = 1.748, Adj. *P* = 5.95e-08.

F   Gene Ontology terms (top) and signaling pathways (bottom) associated with mRNAs differentially expressed between control and *Rx-Dicer* mutants.

G   Heat maps of relative expression levels of DEGs related to apoptosis (top) and proliferation (bottom). Rows correspond to independent biological replicates. Indicated are some example genes.

H   Enrichment plots from GSEA for MSigDB hallmark neurogenesis (NES = 2.98; *P* = 0; Adj. *P* = 0.00045), regulation of cell proliferation (NES = 2.40; *P* = 0; Adj. *P* = 0.0033), positive regulation of cell adhesion (NES = 2.06; *P* = 0.0058; Adj. *P* = 0.013), regulation of response to stress (NES = 2.50; *P* = 0; Adj. *P* = 0.0023), positive regulation of cell death (NES = 1.84; *P* = 0.0179; Adj. *P* = 0.0326), and apoptotic signaling pathway (NES = 1.59; p = 0.047; Adj. *P* = 0.084).

I, J   ISH stains of *Irs2* mRNA, and qPCR for *Irs2* mRNA and *let-7-5p* miRNA, in the rostral telencephalon of control and *Rx-Dicer* mutant embryos. Dashed line indicates basal border, and dotted line indicates apical surface. Arrowheads indicate area with the greatest increase in *Irs2* expression. BG, basal ganglia; H, hippocampus; NCx, neocortex; OB, olfactory bulb; Spt, septum. Histograms represent mean ± SEM (logarithmic scale); symbols in plots indicate values for individual embryos; *t*-test, \**P* < 0.05, \*\**P* < 0.01. N = 6–9 replicates per group. Scale bar, 100 μm.

K   Working model of two potential genetic mechanisms leading to rosette formation in *Dicer* mutants: red, direct effect of increased p53 pathway activity upon loss of *let-7*; green, *Irs2* expression is massively increased by the combined effect of *let-7* loss and increased p53-related cell stress signals, which enhances proliferation and decreases cell adhesion, leading to rosettes.

Taken together, our findings demonstrated that multiple miRNAs and protein-coding genes are deregulated in the rostral telencephalon of E11.5 *Rx-Dicer* mouse embryos. These included members of the *let-7* family of miRNAs, the p53 pro-apoptotic pathway and pathways promoting cell proliferation and adhesion, suggesting their potential relevance in the formation of proliferative rosettes observed in these mutants (Fig 4K). We further hypothesized that high p53-mediated apoptosis might release stress signals that increase expression of genes and pathways promoting proliferation, such as *Irs2* (Fig 4K).

### p53 is required for rosette formation in *Dicer* mutant embryos

Our above analyses revealed that the earliest phenotype in *Rx-Dicer* mutant embryos, preceding the formation of rosettes, was the occurrence of massive cell death starting at E11.5 and that at this stage there was a significant increase in expression of genes related to the p53 signaling cascade, key for apoptosis. In agreement with these findings, we observed a severe downregulation of expression of *let7* miRNAs (two- to fourfold decrease; Fig 4A and B), which have been suggested to repress p53 signaling in a feedback loop manner (Hau *et al*, 2012). So, next we analyzed the potential involvement of p53 signaling in the formation of rosettes in *Rx-Dicer* mutants. Under normal conditions, p53 is expressed at low levels as an inactive peptide, which upon DNA damage undergoes phosphorylation, becoming biologically active (Joerger & Fersht, 2016). Immunostains against phospho-p53 revealed a dramatic increase in positive cells in the rostral telencephalon of *Rx-Dicer* mutants, as compared to controls (Fig 5A–C). This supported the notion that overactivation of p53 might underlie the high levels of apoptosis in these mutants (Siliciano *et al*, 1997). To test this possibility, we conditionally knocked-out *p53* from our *Rx-Cre;Dicer^{F/F}* mutant mice by crossing them with a *p53* floxed mouse line (*p53^{F/F}*) (Marino *et al*, 2000). This resulted in *Rx-Cre;Dicer^{F/F};p53^{F/F}* mice (*Rx-Dicer-p53* double mutants from hereon), deficient in both *Dicer* and *p53* upon *Rx*-driven Cre recombination. 60% of *Rx-Dicer-p53* double mutant embryos displayed a

prominent 35% reduction in apoptosis compared with *Rx-Dicer* single mutants at E12.5 (Fig 5D–H), which was further reduced down to control levels in the remaining 40% of mutant embryos (Fig 5G and H).

Next, we investigated whether the loss of apoptosis in the absence of Dicer, as in *Rx-Dicer-p53* double mutants, might be sufficient to suppress the disruption of apical junctions and the neuronal ectopias observed in *Rx-Dicer* single mutants. Immunostains for Tuj1 and Par3 showed a near-complete rescue of these defects in *Rx-Dicer-p53* double mutants (Fig 5I–L). Given the association of these defects with the formation of rosettes, our observations suggested that the loss of *p53* might also be sufficient to rescue the formation of rosettes in *Rx-Dicer* mutants. Analysis of E17.5 *Rx-Dicer-p53* double mutant embryos showed that the vast majority developed completely normal OB and rostral telencephalon, with no rosettes (eight out of nine embryos), as opposed to single *Rx-Dicer* mutants (Fig 5M–O). Altogether, these results indicated that at early stages of telencephalic development the absence of miRNAs leads to overactivation of the *p53* signaling pathway, which in turn leads to the abnormal development of the rostral telencephalon and the formation of rosettes.

### High levels of *Irs2* drive rosette formation without apoptosis

*Rx-Dicer* mutants displayed overactivation of the p53 signaling, which resulted in high levels of apoptosis and rosette formation without a negative impact on progenitor proliferation. In fact, progenitor cells within rosettes were highly proliferative with an increased rate of cell cycle re-entry compared with controls (Fig 2H). Signals inducing apoptosis, and stress signals released by apoptotic cells, are also known to upregulate genes that promote cell proliferation directly or indirectly (Evan *et al*, 1995; Udelhoven *et al*, 2010). Based on this, we screened our list of DEGs in *Rx-Dicer* mutants to identify genes that promote cell proliferation, are known to be upregulated by stress signals, and are target of *let-7* miRNAs. We identified a single gene fulfilling these three conditions: *insulin receptor substrate 2* (*Irs2*). First, our transcriptomic analyses had

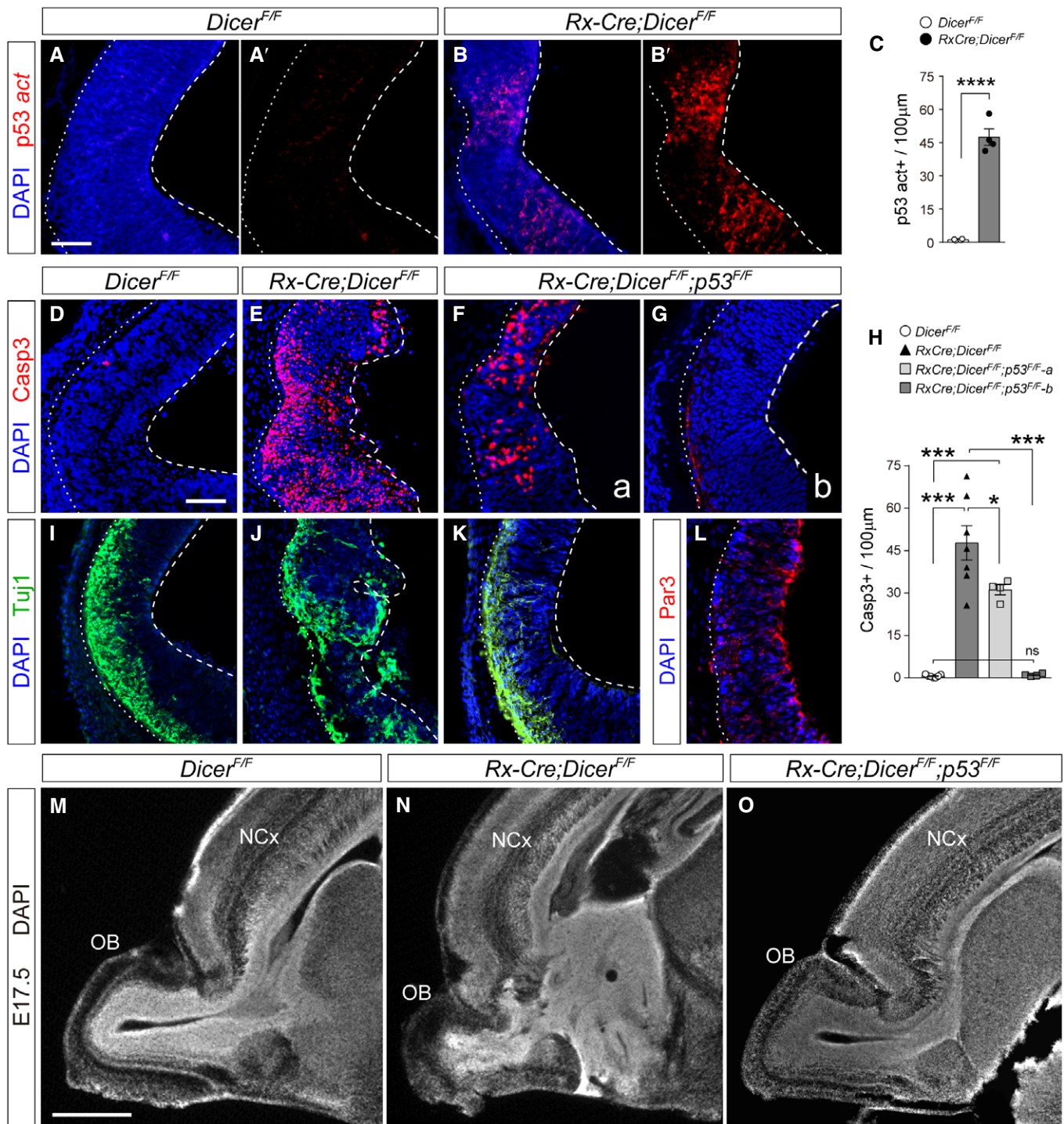

**Figure 5. Loss of *p53* in *Dicer* mutants blocks the formation of rosettes.**

A–C    Immunostains for activated p53 (red) in the rostral telencephalon of control and *Rx-Dicer* mutant embryos at E12.5, and quantification of positive cells (C). Dashed lines indicate apical side, and dotted lines indicate basal side. $N = 4$ replicates per genotype. Scale bar, 50 μm.

D–L    Rostral telencephalon of E12.5 embryos of the indicated genotypes immunostained for the detection of apoptotic cells (Casp3), neurons (Tuj1), and the apical complex protein Par3, and quantification of cells positive for Casp3 (H). $N = 4$–7 replicates per genotype. Scale bar, 50 μm.

M–O    Sagittal sections through the rostral telencephalon of E17.5 embryos of the indicated genotypes. Note the rosettes and general disorganization in *Rx-Dicer* single mutants, and complete rescue of this phenotype in *Rx-Dicer-p53* double mutants. Images in panels M, N are re-used from Fig 2A. NCx, neocortex; OB, olfactory bulb. Scale bar, 1 mm.

Data information: Data in histograms are mean ± SEM, symbols indicate values for individual embryos; *t*-test; ns, not significant; *$P < 0.05$, ***$P < 0.001$, ****$P < 0.0001$.

revealed that *Irs2* levels are increased in the rostral telencephalon of E11.5 *Rx-Dicer* mutant embryos compared with controls, which we independently confirmed by *in situ* hybridization and qPCR (Fig 4E, G, I and J). Moreover, among the most significant signaling pathways related to DEGs in *Rx-Dicer* mutants is the non-alcoholic fatty liver disease pathway, headed by *Irs2* (Fig 4F, Table EV8). Second, previous studies demonstrated that expression of *Irs2* is directly downregulated by *let-7* (Frost & Olson, 2011; Zhu *et al*, 2011). Third, our GSEA identified *regulation of response to stress* among the most significant terms (Fig 4H).

To determine whether the phenotype observed in *Rx-Dicer* mutants, and rescued upon loss of p53, was the result of upregulation of *Irs2*, we next performed acute overexpression experiments by *in utero* electroporation in E12.5 wild-type embryos, where all other developmental parameters were otherwise normal. Two days after overexpression of *Irs2* in the rostral telencephalon, we observed the presence of ectopic neurons in the VZ and proliferative rosettes in the telencephalic parenchyma (Fig 6A–E). These rosettes exhibited the characteristic features found also in *Rx-Dicer* mutants, including a small central lumen limited by a Par3$^+$ AJ belt and a circle of apical mitoses, and then a band of basal mitoses, and surrounded externally by Tuj1$^+$ neurons (Fig 6F–H). The presence of the AJ belt inside these rosettes in electroporated wild-type embryos indicated that they formed by invagination of the VZ, as in *Rx-Dicer* mutants. Five days after electroporation (E17.5), rosettes remained in the white matter (Fig 6D), largely reminiscent of *Rx-Dicer* mutants (Fig 2D).

The above analyses of Rx-Dicer mutants showed that hyperproliferative rosettes affected the neocortex only in its most rostral region, not in the parietal or caudal parts (Figs 2D, and EV3A and B). Because Cre recombination and loss of miRNAs in these mutants were highly regionalized in a similar pattern (Figs EV1 and 2), it was unclear whether this might be the cause of regionalized rosette formation. To elucidate whether other neocortical regions are susceptible to form rosettes following high *Irs2* expression, we performed *in utero* electroporation of wild-type embryos at E12.5 to overexpress *Irs2* in a very large extension of the neocortical primordium starting from its rostral end, next to the OB (Fig EV3C). Analysis at E14.5 revealed the formation of rosettes in the most rostral aspect of the neocortex, as expected.

However, this defect was sharply interrupted, with the remaining more caudal regions seemingly perfectly normal in spite of similar levels of electroporation (Fig EV3C). This showed that not all regions of the cortical neuroepithelium are susceptible to high *Irs2* levels in the formation of rosettes, suggesting the existence of different genetic mechanisms to maintain cellular homeostasis across the developing telencephalon.

Our findings so far demonstrated that the rosette phenotype of *Rx-Dicer* mutants requires overactivation of the p53 pathway and that it is phenocopied in normal embryos by overexpression of *Irs2* alone. This was in full agreement with our working model, where both the loss of *let-7* and the release of stress signals upon p53-mediated apoptosis may increase *Irs2* expression in *Rx-Dicer* mutants, and this may lead to formation of rosettes (Fig 4K). Because rosettes in *Rx-Dicer* mutants were always linked to increased proliferation, impaired adherens junctions, and massive apoptosis, it was still unclear whether the formation of rosettes by *Irs2* overexpression in wild-type embryos involved solely increased proliferation and perturbed cell adhesion, or also massive cell death. To test this, we again overexpressed *Irs2* by *in utero* electroporation of wild-type embryos at E12.5 (where all other developmental parameters were normal) and analyzed the effects of this overexpression on apoptosis 1 day later, at E13.5, a time when rosettes were forming following this manipulation (Fig 6C). In contrast to controls, *Irs2*-overexpressing embryos showed a massive displacement of Tuj1$^+$ neurons from CP to the ventricular surface, coupled to the displacement of apical mitoses to basal positions, seemingly forming nascent rosettes (Fig 6I–K'). Importantly, however, the abundance of Casp3$^+$ apoptotic cells among cortical progenitors in the VZ was negligible, similar to control GFP-electroporated embryos (Fig 6M) and completely different than in *Rx-Dicer* mutants (Fig 1B and C). The only significant increase in Casp3$^+$ cells was at the CP (Fig 6J–M), which was not part of rosettes, and nevertheless was one order of magnitude lower than in *Rx-Dicer* mutants at the age of rosette onset (Fig 1E). Taken together, our experiments in wild-type embryos, where expression of miRNAs and levels of apoptosis were normal, demonstrated that increased expression of *Irs2* alone was in itself sufficient to induce the formation of rosettes, and this did not involve the massive apoptosis of progenitor cells.

**Figure 6. Rosettes form by upregulation of *Irs2* expression.**

A Relative *Irs2* mRNA levels in HEK cells upon transfection with *Gfp*- or *Irs2*-encoding plasmids. Mean ± SEM; *t*-test, ****$P < 0.0001$. $N = 6$ replicates per group.

B–D Distribution of neurons (Tuj1, red), GFP (green), and mitoses (PH3, white) in the rostral telencephalon of wild-type E14.5 or E17.5 mouse embryos, as indicated, electroporated at E12.5 with the indicated plasmid combinations. Dashed line indicates the apical surface. Insets show the same field of view, with location and extent of electroporations. Note the overlap of rosettes (arrows) with ectopic neurons at the apical surface (arrowheads) and increased basal mitoses, upon overexpression of *Irs2* (C). Asterisk indicates the absence of malformation in nearby cortex. At E17.5 (D), GFP$^+$ rosettes remain in deep cortical layers (arrowheads).

E–H Details of rosettes (large arrows in E; dashed lines in F–H) from *Irs2* + *Gfp*-overexpressing E14.5 embryos immunostained as indicated. Small arrows indicate apical mitoses, arrowheads indicate basal mitoses (F, G) and Par3$^+$ apical surface at the center of rosettes (H), dotted lines indicate apical surface, and dashed lines indicate basal side of rosettes. Inset in (H) is at the same scale.

I–M Distribution of neurons (Tuj1, red), GFP (green), and mitotic (PH3, white) or apoptotic (Casp3, white) cells in the rostral telencephalon of wild-type E13.5 embryos electroporated at E12.5 with the indicated plasmids, and quantification of Casp3$^+$ cells (M). Dashed line indicates apical surface, and dotted lines indicate borders of the cortical plate (CP). Nascent rosettes (arrows) are next to ectopic ventricular neurons (arrowheads in K') and basal mitoses, without significant apoptosis in VZ (L', M). Mean ± SEM; *t*-test; ns, not significant, *$P < 0.05$. $N = 3–6$ replicates per group.

N Distribution of neurons (Tuj1, red) in the rostral telencephalon of wild-type E14.5 embryos electroporated at E12.5 with a combination of plasmids encoding *Irs2*, *let-7a*, *let-7b* and *let-7c*, and *Gfp*. Dashed line indicates apical surface. Formation of rosettes in *Irs2*-expressing embryos is rescued by overexpression of *let-7*. Inset shows the same field of view, with location and extent of electroporation.

Data information: Scale bars, 150 μm (B–D, N), 50 μm (E–H), 100 μm (I–L).

 

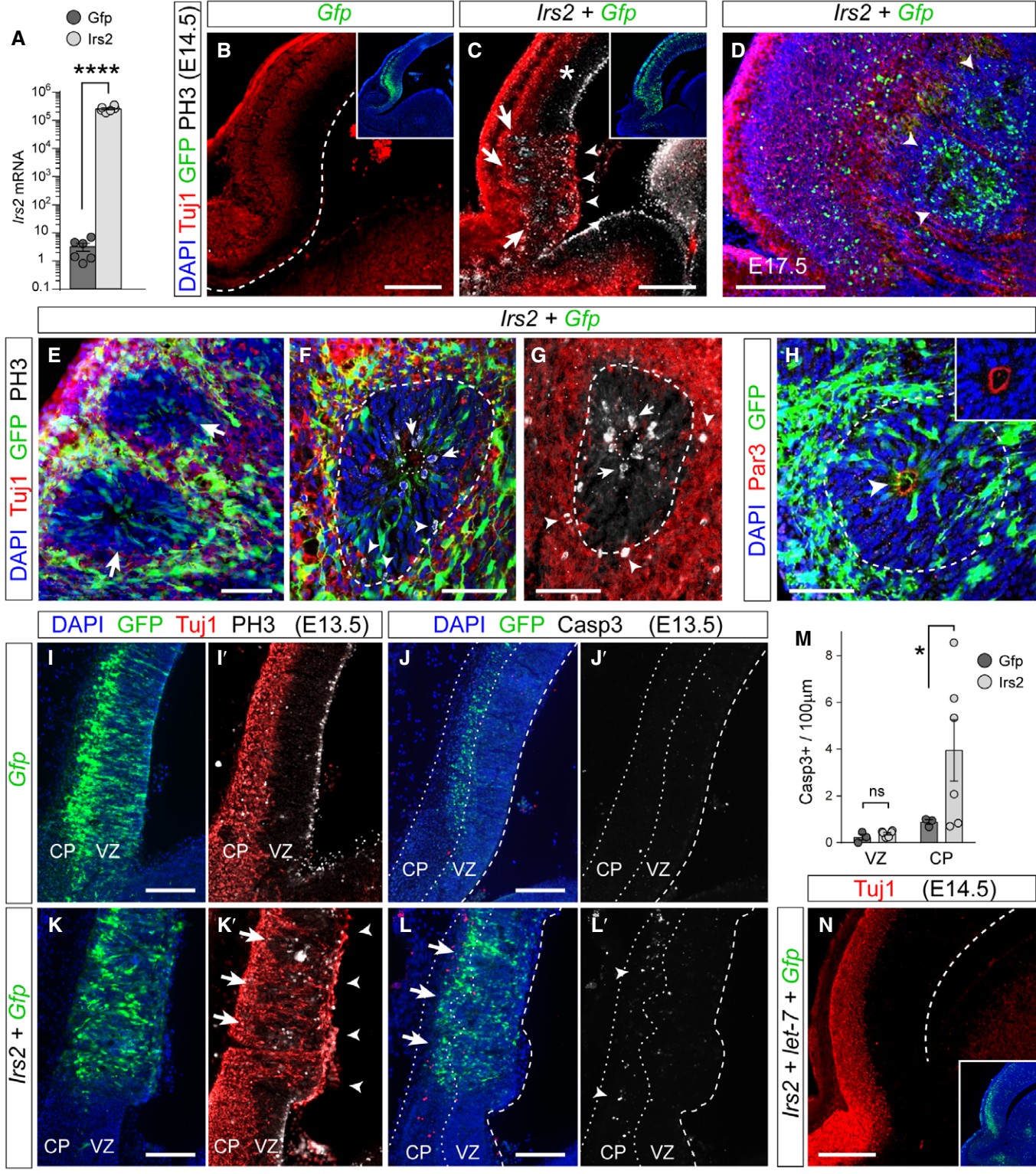

**Figure 6.**

### let-7 miRNAs block the Irs2-driven formation of rosettes

Our results demonstrated that the loss of cellular homeostasis and formation of rosettes in the telencephalic neuroepithelium require

high levels of Irs2, but not massive apoptosis, supporting the notion that this phenotype in *Rx-Dicer* mutants emerges from their increased *Irs2* levels. Because *Irs2* is a direct target of, and downregulated by, *let-7* miRNAs (Frost & Olson, 2011; Zhu *et al*, 2011), and

these are downregulated in *Rx-Dicer* mutants, increased levels of *Irs2,* and formation of rosettes in these mutants might be the consequence of low *let-7* levels. To directly test this possibility and confirm the functional interaction between *let-7* and Irs2 in the formation of rosettes, we co-electroporated wild-type embryos with plasmids encoding *Irs2* together with a mixture of three mature *let-7* miRNAs (*let-7a-5p, let-7b-5p,* and *let-7c-5p*) that target the *Irs2*

coding sequence. The addition of these *let-7* miRNAs was sufficient to rescue the malformations in a majority of embryos overexpressing *Irs2* (Figs 6N and 7A).

Next, we studied whether the loss of endogenous, physiological levels of *let-7* in wild-type embryos is sufficient in itself to recapitulate the rosette phenotype observed in *Rx-Dicer* mutants, and upon experimental overexpression of *Irs2* in control animals. To perform

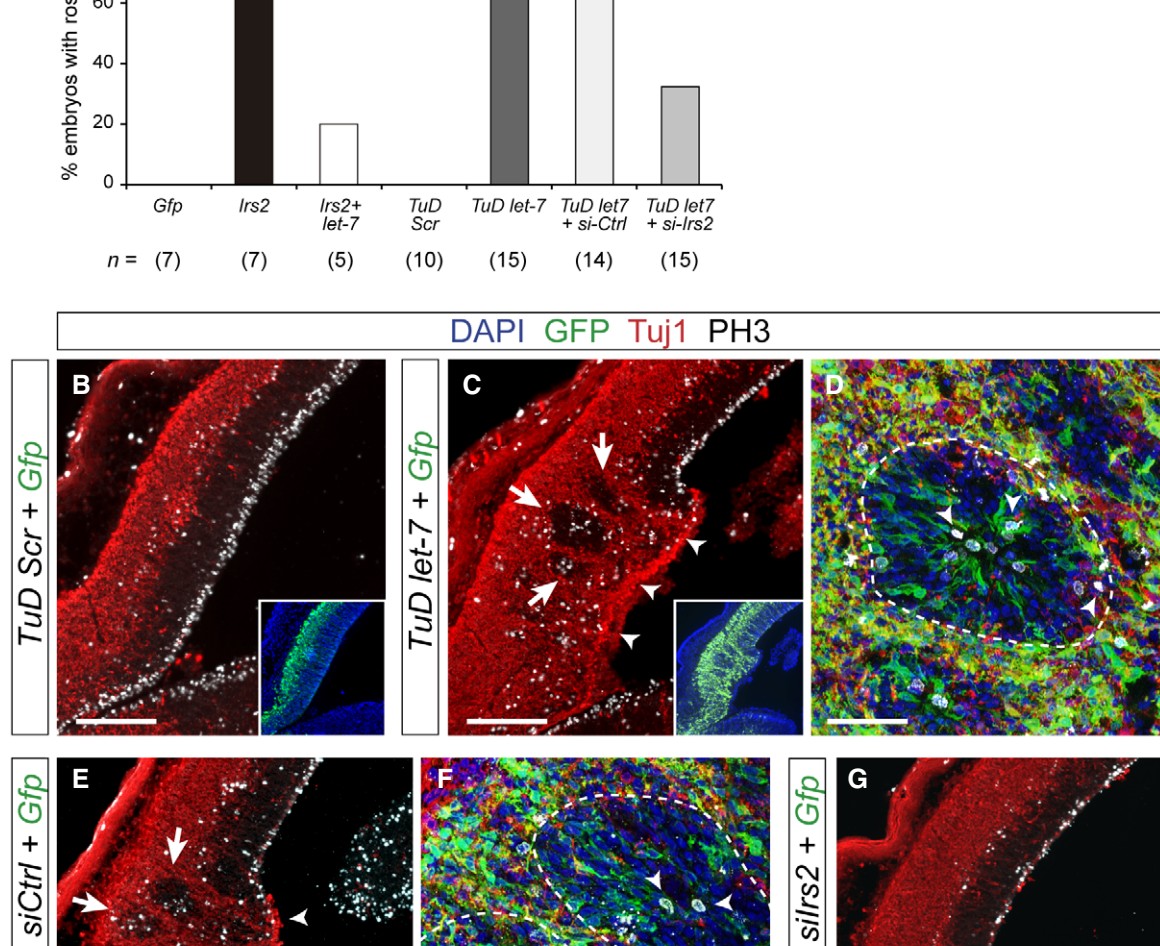

**Figure 7. Rosettes form by loss of *let-7* and upregulation of *Irs2* expression.**

A Frequency of rosette formation in the experimental conditions indicated. *n*, number of embryos per condition.

B–G Distribution of neurons (Tuj1, red), GFP (green), and mitoses (PH3, white) in the rostral telencephalon of wild-type E14.5 mouse embryos electroporated at E12.5 with the indicated plasmid combinations. Inset shows the same field of view, with location and extent of electroporations. In (C) and (E), arrows indicate rosettes and arrowheads indicate neuronal ventricular ectopias. (D) and (F) are high magnification details of individual rosettes (dashed lines), where arrowheads indicate apical and basal mitoses. Loss of endogenous *let-7* drives the formation of rosettes and ventricular neuronal ectopias (C), which is rescued with the additional loss of *Irs2* (G).

Data information: Scale bars, 150 µm (B, C, E, G), 50 µm (D, F).

loss of function of *let-7,* we generated sequence-specific Tough Decoys (TuD *let-7*), which decreased more than 80% the endogenous levels of *let-7* in HEK cells (Fig EV4A). Two days after *in utero* electroporation of *TuD let-7* in the rostral telencephalon of E12.5 wild-type embryos, distinct proliferative rosettes and neuronal ectopias had formed in 73% of cases (similar to *Irs2* overexpression), but not in control *TuD-Scr* electroporated embryos (Fig 7A–D). These rosettes were very similar to those observed in *Rx-Dicer* mutants and upon *Irs2* overexpression, consistent on a concentric array of radial glia-like progenitor cells with apical mitoses in the central lumen, basal mitoses in the periphery, and externally surrounded by Tuj1$^+$ neurons (Fig 7C and D). This result, together with our transcriptomic analyses in *Rx-Dicer* mutants, suggested that formation of hyperproliferative rosettes is due to increased levels of *Irs2* as a result of the loss of *let-7*. Consistent with this notion, expression of *TuD let-7* in the rostral cortex led to a detectable 40% increase in endogenous Irs2 protein (Fig EV4B and C), demonstrating that endogenous *let-7* reduces Irs2 protein levels. To test our hypothesis functionally, we performed rescue experiments where wild-type embryos were co-electroporated with *TuD let-7*, and with siRNAs against *Irs2* for loss of function. Whereas control siRNAs had null influence on the formation of rosettes by *TuD let-7*, which continued to be observed in 71% of embryos, siRNAs against *Irs2* were sufficient to completely suppress this phenotype in a majority of *TuD let-7* embryos (observed in only 33% of cases; Fig 7A and E–G). Altogether, our results demonstrated that the formation of rosettes in *Rx-Dicer* mutants is the result of the upregulation of *Irs2* caused by the loss of *let-7*.

Finally, we tested whether the novel role of Irs2 in promoting the proliferation of murine cortical progenitor cells is conserved in the developing human cortex. We generated dorsal cerebral organoids from healthy human induced pluripotent stem cells (hiPSCs), and at culture day 40, we electroporated *Irs2*-encoding plasmids (Fig 8A). Seven days later, the proportion of GFP$^+$ cells positive for Ki67 was significantly higher in *Irs2*-overexpressing organoids than in controls (Fig 8B–D), which indicated greater progenitor amplification versus differentiation, as observed in the rosettes of *Rx-Dicer* mutant embryos (Fig 2H). A similar effect was obtained upon electroporation of cerebral organoids with *TuD-let7*, compared with controls expressing *TuD-Scr* (Fig 8E–G). Together, these results demonstrated that endogenous *let-7* miRNAs also limit the amplification of human cortical progenitor cells, boosted by Irs2 overexpression. Effects on rosette formation were not discernible in cerebral organoids because these are already constituted by multiple small ventricles, as opposed to the single large ventricle of the embryonic telencephalon.

In summary, our results demonstrate that in the early telencephalic neuroepithelium *let-7* miRNAs limit the expression levels of *Irs2* and components of the *p53* signaling pathway, which is essential to maintain the cellular homeostasis of the proliferative telencephalic neuroepithelium (Fig 8H). In *Rx-Dicer* mutant embryos, the loss of *let-7* miRNAs de-represses *Irs2* expression and p53 pathway signaling, dramatically augmenting apoptosis and, potentially, further increasing *Irs2* expression. The resulting high levels of Irs2 cause an imbalanced overgrowth of the neuroepithelium, with progenitor cell overproliferation and loss of apical adherens junctions, leading to a loss of structural integrity and the formation of hyperproliferative rosettes (Fig 8H).

# Discussion

microRNAs are widespread regulators of gene expression in development and disease (Bartel, 2018), but the extent of their contribution to early fetal brain development remains elusive, particularly regarding the telencephalon. By using the early recombining *Rx-Cre; Dicer$^{F/F}$* mouse line, here we demonstrate that miRNAs are important for telencephalic development much earlier, and at a much greater extent, than previously suspected. *Rx-Dicer* mutant embryos display severe deregulation of gene expression and altered pathway signaling at very early stages of telencephalic development, starting at E11.5 when we find overactivation of the p53 pathway. This is followed by massive apoptosis of progenitor cells in multiple telencephalic regions. In the rostral telencephalon, massive cell death is combined with loss of apical–basal polarity and increased proliferation of progenitor cells, leading to the formation of hyperproliferative rosettes. This complex phenotype is caused by the loss of *let-7* miRNAs and a concomitant increase in *Irs2* expression levels, a direct target of *let-7* (Zhu *et al*, 2011).

Maturation of the vast majority of mammalian miRNAs depends on Dicer, an enzyme that processes pre-miRNAs into mature, socalled Dicer-dependent, miRNAs. Maternal miRNAs are essential for zygotic development (Tang *et al*, 2007), and zygotic loss of *Dicer* is lethal at the epiblast stage (Bernstein *et al*, 2003). This evidences the fundamental importance of miRNAs in mammalian development from the earliest stages, and the need of using conditional deletion of *Dicer* to study their functions. Previous analyses using early *Cre* drivers to conditionally knockout *Dicer*, such as *Emx1* and *Nestin*, concluded that miRNAs have a modest role in regulating neural progenitor cells and the embryonic development of the telencephalon, beyond preventing massive apoptosis (De Pietri Tonelli *et al*, 2008; McLoughlin *et al*, 2012; Saurat *et al*, 2013). However, other studies show that the proliferation, reprogramming, and differentiation of stem cells, including cancer stem cells, are under tight regulation by various miRNAs (Chakraborty *et al*, 2016; Li *et al*, 2017). Here, by using the *Rx-Cre* driver, expressed since E7.5, we have uncovered that miRNAs are in fact very important in telencephalic development at much earlier stages than previously suspected and that the defects caused by their absence go much beyond apoptosis. Our findings show that the early loss of miRNAs not only induces high levels of apoptosis, as previously shown, but also it changes significantly the proliferative activity and lineage dynamics of neural progenitor cells, switching from self-renewal and neurogenesis to self-amplification and expansion. In addition, we find that miRNAs are required to maintain the integrity of the apical adherens junction belt of the telencephalic neuroepithelium, in the absence of which the polarity of cortical germinal zones is altered and newborn neurons accumulate within the VZ. These are completely novel and unexpected roles of miRNAs that extend the palette of known functions during embryogenesis (Bartel, 2018) and, most importantly, uncover novel critical roles in regulating germinal layer homeostasis, neural stem cell dynamics, and neurogenesis in the developing telencephalon, not previously recognized (De Pietri Tonelli *et al*, 2008).

Previous *in vivo* and *in vitro* studies demonstrated that *Dicer* mutants typically exhibit high levels of apoptosis in the developing neocortex at late stages, concomitant with decreased cortical thickness (Mott *et al*, 2007; Raver-Shapira *et al*, 2007; Davis *et al*, 2008;

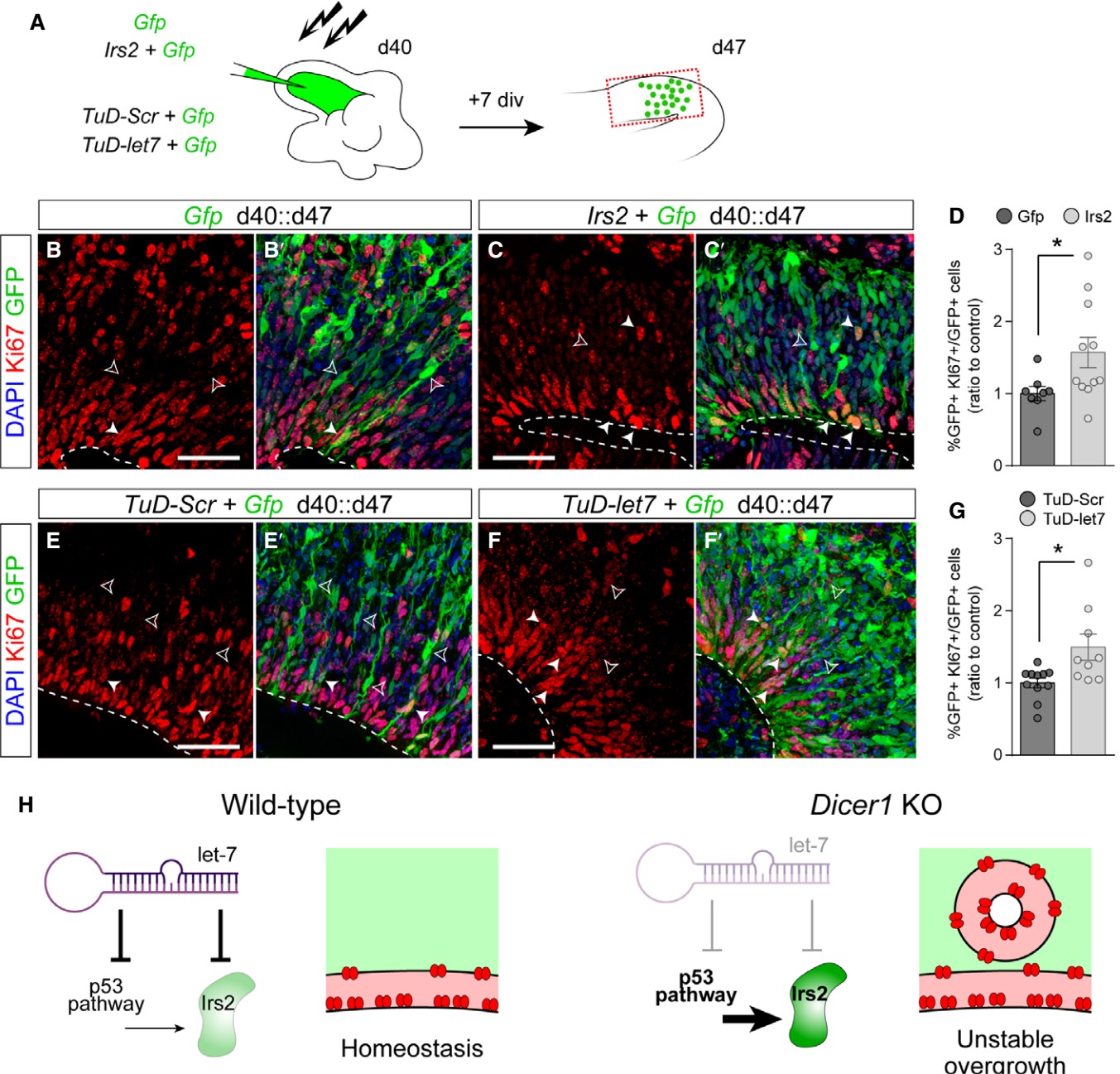

**Figure 8.  *Irs2* and *let-7* regulate progenitor cell proliferation in human cerebral organoids.**

A     Schema of experimental design. At 40 days in culture, human cerebral organoids were electroporated with one of the indicated DNA plasmid combinations, followed by seven additional days in culture prior to analysis.

B–G   Stains and quantifications of human cerebral organoids electroporated with the indicated construct combinations, to reveal the frequency of electroporated cells remaining as progenitors (%GFP⁺Ki67⁺/GFP⁺). Histograms represent mean ± SEM; symbols in plots indicate values for individual organoids; chi-square test, *$P < 0.05$. N = 8–11 replicates per group. Dashed lines indicate ventricular surface, solid arrowheads indicate Ki67⁺/GFP⁺ cells, and open arrowheads indicate Ki67⁻/GFP⁺ cells. Scale bars, 50 μm.

H     Schematic drawing of conclusions from this study. *let-7* expressed in the early telencephalic neuroepithelium represses the expression of p53 and Irs2, which maintains the cellular homeostasis and epithelial integrity. Loss of *let-7*, as in *Dicer1* mutants, de-represses the expression of *Irs2* and of p53 pathway components, which further increase *Irs2* levels. High Irs2 enhances progenitor cell proliferation and loss of apical adhesion, altering neuroepithelial homeostasis and integrity, and leading to the formation of hyperproliferative rosettes.

De Pietri Tonelli *et al*, 2008). Although we also observed massive levels of apoptosis in the telencephalic primordium of *Rx-Dicer* mutants, these were mostly restricted to the rostral and ventral telencephalon at early stages, while the neocortex seemed largely unaffected (Fig 1A). Similar to previous Dicer mutants, we did observe a reduction in telencephalic volume in *Rx-Dicer* mutants at E17.5 compared with littermates (Fig EV5A and B), which included a reduction in cerebral cortex thickness affecting the CP but not the germinal layer VZ/SVZ (Fig EV5C and D). Cortical thinning was much more pronounced at E12.5 and E14.5 (Fig EV5E and F), but it did not alter the normal laminar arrangement of superficial (Tbr1$^+$) and deep layer (Ctip2$^+$) neurons (Fig EV5C). The selective thinning of cortical neuronal layers was similar to the phenotype reported for *Emx1-Cre;Dicer$^{F/F}$* mutants, where this in fact resulted from progenitor cell apoptosis, which increased from E12.5 to E14.5 (De Pietri Tonelli *et al*, 2008). Although we also observed a greater occurrence of apoptosis in *Rx-Dicer* mutants, this was highest at E11.5 and decreased down to normal levels already by E13.5 (Fig EV5G and H). Apoptosis was accompanied by reductions in Tbr2$^+$ cells (a marker of IPCs and newborn neurons) in *Rx-Dicer* mutants (Fig EV5I and J). This developmental decline in both cell death and Tbr2$^+$ cells was inconsistent with analyses of *Emx1-Cre;Dicer$^{F/F}$* mutants, where apoptotic cells continued increasing by E14.5 and Tbr2$^+$ cells were unaltered (De Pietri Tonelli *et al*, 2008). Strikingly, analyses of mitotic activity via PH3 stains showed only sporadic and minor differences in density of apical and basal mitoses (Fig EV5K and L). This indicated that deficits in cortical progenitors were largely specific to neurogenesis (Tbr2$^+$ cells), whereas proliferation *per se* was not compromised, indicating a potential imbalance in progenitor cell fate from neurogenesis to self-renewal. Hence, Dicer and Dicer-dependent miRNAs are necessary in the early embryonic murine cerebral cortex to prevent death of progenitor cells and promote neurogenesis, as shown previously using other *Dicer* mutants. However, our findings demonstrate that the necessity of Dicer in cortical development begins at earlier stages than previously reported, and that these early defects in *Rx-Dicer* mutants are transient, in contrast to previous studies using *Emx1-Cre* mice. Given the low efficiency of Cre recombination, and the persistence of miRNAs, in the dorsal neocortex of *Rx-Dicer* mutants until late embryonic stages (Figs EV1 and EV2), our results likely underestimated the physiological relevance of miRNAs at mid-late cortical development, potentially explaining the discrepancies between our current findings and those of previous reports using other Cre driver mouse lines.

Our findings in mouse embryos are consistent with mounting data showing that deregulation of miRNAs is related to many types of cancer (Di Leva *et al*, 2014). Interestingly, some of the pediatric cancers include formation of proliferative rosettes (Korshunov *et al*, 2010; Sturm *et al*, 2016), unique structures with features typical of the embryonic germinal zones that we find massively produced in *Rx-Dicer* mutant mouse embryos. Our transcriptomic analyses and experimental manipulations of early developing embryos uncover the genetic mechanism underlying this phenotype. We show that the absence of *Dicer* during early embryonic development of the telencephalon has a particularly important impact on *let-7*. This is a family of miRNAs highly expressed during development that target and directly repress the expression of multiple developmentally relevant genes. Accordingly, in *Rx-Dicer* mutant embryos the most

highly upregulated genes were all direct targets of *let-7*: *Protogenin* (*Prtg*), *Lin28*, followed by a specific enrichment in *p53* pathway-related genes (Fig 4) (Moss & Tang, 2003; La Torre *et al*, 2013; Li *et al*, 2014; Subramanian *et al*, 2015). In turn, p53 signaling and Lin28 have been shown to reciprocally repress *let-7*, defining two negative feed-back loops (Newman *et al*, 2008; Jones & Lal, 2012; Sun *et al*, 2015; Farzaneh *et al*, 2017). In agreement with such findings in other systems, in the embryonic telencephalon of *Rx-Dicer* mutants we find that the loss of *let-7* is linked to the overactivation of p53 and to p53-driven massive apoptosis. *Prtg* is known to prevent premature apoptosis in the rostral cephalic region (Wang *et al*, 2013). Thus, the increase in *Prtg* expression in the rostral telencephalon of *Rx-Dicer* mutants may interfere in a delayed manner with their massive apoptosis, explaining its short duration between E11.5 and E13.5. Moreover, the formation and expansion of proliferative rosettes correlates with the increase in *Lin28a*, *Prtg,* and its downstream mediator *β1-integrin*, which are known to maintain the neural progenitor state and prevent neuronal differentiation (Wong *et al*, 2010; La Torre *et al*, 2013). In fact, Lin28a is a marker of rosette-forming embryonic tumors (Korshunov *et al*, 2010), suggesting a potential mechanistic link between the formation of rosettes in *Rx-Dicer* mutant embryos and certain types of pediatric brain tumors.

*let-7* is known as a potent tumor suppressor targeting multiple oncogenes (Pobezinsky & Wells, 2018). In addition to regulating *Lin28*, *let-7* miRNAs directly target and repress the expression of *Irs2* (Zhu *et al*, 2011), a positive regulator of cell proliferation upregulated in liver, pancreas, and prostate cancer (Mardilovich & Shaw, 2009; Mardilovich *et al*, 2009). *Irs2* expression is enhanced by stress signals, including metabolites and pro-inflammatory peptides such as those typically released during apoptosis (Udelhoven *et al*, 2010; Zhu *et al*, 2011). In the rostral telencephalon of *Rx-Dicer* mutants, we find increased expression of *Irs2*, which may result from a combination of both the loss of *let-7* and high apoptosis. Importantly, our *in utero* manipulations in wild-type embryos demonstrate that the overexpression of *Irs2* alone is sufficient both to elicit the formation of proliferative rosettes and for their maintenance throughout embryonic development, which occurs in the absence of apoptosis. This phenotype is rescued by overexpressing *let-7*, and replicated by blocking the endogenous physiological levels of *let-7* in wild-type embryos, which we have shown increases the endogenous levels of Irs2 protein. In a majority of cases, the formation of rosettes by blockade of *let-7* expression was completely suppressed by simultaneous blockade of *Irs2* expression, demonstrating their functional link. This result was remarkable given that *let-7* miRNAs target several hundred genes (Table EV3), which further highlights the central role of *Irs2* in the emergence of this phenotype. Intriguingly, although high *Irs2* induces rosette formation in the absence of apoptosis, rosettes in *Rx-Dicer* mutants were rescued by the loss of *p53*, suggesting a role for p53 signaling in the regulation of *Irs2* levels independent from apoptosis. P53 has been shown to negatively regulate *let-7* function, by downregulating both its expression and its binding to AGO (Saleh *et al*, 2011; Hau *et al*, 2012; Subramanian *et al*, 2015; Krell *et al*, 2016); so, a loss of p53 may lead to a net increase in *let-7* function. We have shown that in the rostral telencephalon of *Rx-Dicer* mutant embryos there is some remaining amount of *let-7* miRNA (Fig 4J). Hence, in double *Rx-Dicer-p53* mutants, the loss of p53 may allow partially higher levels of active

*let-7*, which combined with the already remaining *let-7* expression levels prior to *p53* loss, may be sufficient to limit *Irs2* levels and prevent rosette formation. Altogether, these results point at Irs2 as a key player in the destabilization of the embryonic neuroepithelium and the formation of proliferative rosettes, and to *let-7* miRNAs as key regulators of *Irs2* expression in this context.

Our study demonstrates that maintenance of the cellular homeostasis and structural integrity of the embryonic telencephalic neuroepithelium is under the critical regulation of miRNAs. The early absence of miRNAs leads to transcriptomic deregulation and the pathological formation of neural rosettes, resembling those observed in various types of pediatric cancer. This involves the combination of multiple cellular mechanisms including cell proliferation, apoptosis, and destabilization of apical adherens junctions. These findings support the notion that miRNAs may be used as a strategy for the genetic intervention of pediatric oncogenic disease (Pobezinsky & Wells, 2018).

# Materials and Methods

## Animals

Wild-type and Rx3-Cre mice (Swindell *et al*, 2006) were maintained in a CD1 background. Mice carrying floxed alleles for Dicer (Dicer1tm1Bdh/J) (Harfe *et al*, 2005), p53 (B6;129P2-Trp53tm1Brn/J) (Marino *et al*, 2000) (generous gift of Anton Berns and Maria Blasco) and the tdTomato reporter line (B6;129S6-Gt(ROSA) 26Sortm9(CAG-tdTomato)Hze/J) (Madisen *et al*, 2010) were maintained in C57BL/6 background. The day of vaginal plug was considered embryonic day (E) 0.5, and embryos were used of either gender. Mice were kept on a 12:12-h daylight cycle at the Animal Facility of the Instituto de Neurociencias of Alicante, and experimental procedures were performed in compliance with institutional Spanish and European regulations (CSIC Ethics Committee).

## *In situ* hybridization and immunohistochemistry

For the detection of coding mRNAs, sense and anti-sense cRNA probes were synthesized and labeled with digoxigenin using the DIG RNA Labeling Kit (SP6/T7; Roche, 11175025910) according to the manufacturer's instructions. For the detection of *Dicer* exons 22 and 23, we used a specific probe from a fragment cloned with the following primers: forward, CCAAGCCCAGCAATGAATGT; reverse: CCAAAATCGCATCTCCCAGG. For the detection of *Irs2*, we used probes from a fragment cloned with the following primers: forward, AGACCCTAAGCTACTCCCCA; reverse, GCTGTAAGGAGGAAGGGGAA. *In situ* hybridization (ISH) was performed on frozen brain sections as described elsewhere (De Juan Romero *et al*, 2015). For the detection of mature miRNAs, we used miRCURY™ LNA™ microRNA ISH Detection Probes (Qiagen) using locked-nucleic-acid-modified (LNA) probes, previously shown to be specific for mature miRNAs but not their precursors (Kloosterman *et al*, 2006).

For immunohistochemistry, frozen or vibratome brain sections were incubated with primary antibodies overnight, followed by appropriate fluorescently conjugated secondary antibodies and counterstained with 4′,6-diamidino-2-phenylindole (DAPI; Sigma, D9542). Primary antibodies used were against Arl13b (1:500,

Abcam, ab83879), β-catenin (1:2,000, Sigma, C2206), BrdU (1:500, Abcam, ab6326), cleaved Caspase 3 (1:150, Werfen, 9661), Ctip2 (1:500, Abcam, ab18465), GFP (1:1,000, Aves Lab, GFP-1020), Ki67 (1:500, Abcam, ab15580), phosphohistone H3 (1:1,000, Upstate, 06-570), Tbr1 (1:500, Abcam, ab31940), Tbr2 (1:500, Millipore, ab31940), βIII-tubulin (1:1,000, Covance MMS-435P), p53 phospho S15 (1:500, Abcam, ab1431), Par3 (1:500, Millipore, MABF28), and Pax6 (1:500, Millipore, AB2237). Secondary antibodies were from Vector Lab: biotinylated anti-Rabbit IgG (1:200, BA-1000); from Jackson InmunoResearch: biotinylated Fab anti-Rabbit IgG (1:200, 711-067-003), Alexa 488 anti-chicken IgY (1:200, 703-545-155), Cy2 Streptavidin (1:200, 016-220-084), Cy5 Streptavidin (1:200, 016-170-084); and from Invitrogen: Alexa 488 anti-mouse IgG (1:200, A-21202), Alexa 488 anti-rabbit IgG (1:200, A-21206), Alexa 555 anti-mouse IgG (1:200, A-31570), and Alexa 555 anti-rabbit IgG (1:200, A-31572).

## Bromodeoxyuridine labeling experiments

To identify progenitor cells in S phase, a single dose of BrdU (50 mg/kg body weight) was injected at E17.5, embryos were fixed 30 min later, and the percentage of Ki67$^+$ cells labeled with BrdU was calculated. To calculate cell cycle re-entry, a single dose of BrdU was administrated 24 h prior to embryo fixation, and then, the percentage of BrdU$^+$/Ki67$^+$ cells was calculated.

## RNA sequencing analysis

The rostral-most region of the telencephalic primordium was dissected out from *Dicer*$^{F/F}$ and *Rx-Cre-Dicer*$^{F/F}$ E11.5 embryos, and tissue pieces were immediately frozen. Tissue was then dissociated with Papain (Miltenyi Biotec, 130-092-628), RNA was extracted using Quick RNA Mini Prep (Zymo Research, R1055), and 1 μg of total RNA was used for library preparation and sequencing. RNA integrity was analyzed using Bioanalyzer (Agilent 2100) and RIN values of the samples varied between 8.2 and 8.9 (average of 8.7), indicating high quality for sequencing library construction. Libraries were prepared either with NEB Next Ultra Directional RNA Library Prep Kit (for RNA-seq) or NEB Next Small RNA Library Prep Kit (for miRNA-Seq) and sequenced on Illumina HiSeq 2500 with 100-bp paired-end read RNA-Seq, and 75-bp single-end read Small RNA-Seq. Reads were quality-checked with FASTQC v2.6.14. Small RNA-Seq reads were analyzed using miRDeep2 (Friedlander *et al*, 2008) and mm10 and miRBase (v22) as reference. RNA-Seq output reads were quality-trimmed and filtered for adapters using Trim Galore v0.6.5 and aligned to the genome assembly (GRCm38.p6) using HISAT2 (Kim *et al*, 2015). Gene-level read counts were computed using HTSeq (parameters: -m union –stranded = reverse and Ensembl gene annotations) (Anders *et al*, 2015). Differentially expressed genes were identified using DESeq2 (Love *et al*, 2014). The miRWalk v3.0 algorithm was used to identify the putative target genes of differentially expressed miRNAs in *Rx-Cre-Dicer*$^{F/F}$ (Dweep & Gretz, 2015). Putative targets were filtered using miRTarBase, to retain only validated targets. Gene Ontology (GO) functional enrichment analysis for DEGs and Kyoto Encyclopedia of Genes and Genomes (KEGG) pathway enrichment analysis were performed using the DAVID online tool v6.8 (da Huang *et al*, 2009) and clusterProfiler (Yu *et al*, 2012). Gene set enrichment analysis method was also used for enrichment

analysis of the DEGs in *Rx-Cre-Dicer^F/F* (Subramanian *et al*, 2005). Raw and processed RNA-Seq data for both RNA-Seq and small RNA-Seq are available at GEO accession GSE151150.

## Quantitative real-time PCR

For RNA extraction, brains were dissected in cold RNase-free medium into small tissue blocks and immediately frozen in liquid nitrogen. Total RNA was extracted using RNeasy Mini Kit (Qiagen) followed by treatment with RNase-Free DNase Set (Qiagen). Template cDNA was generated using Maxima First-Strand cDNA Synthesis Kit for quantitative real-time PCR (qRT–PCR; Thermo Fisher). Quantitative PCR was performed in a StepOnePlus Real-Time PCR System (Applied Biosystems, Foster City, CA, USA) using the MicroAmp Fast 96-well reaction plate. A master mix was prepared for each primer set containing TaqMan™ Gene Expression Master Mix (Applied Biosystems, Ref: 4369016), primers, and template cDNA. Expression of *Irs2* mRNA was determined using the TaqMan assay (Applied Biosystems): Irs2 Hs00275843_s1. 18S Hs99999901_s1 ribosomal RNA was used as loading control. All reactions were performed in technical duplicates. The amplification efficiency for each primer pair and the cycle threshold ($C_t$) were determined automatically by StepOne v2.2.2 (Applied Biosystems) or QuantStudio 3 (Thermo Fisher Scientific) softwares. *Irs2* transcript levels were represented relative to the 18S signal adjusting for the variability in cDNA library preparation.

For miRNA, small RNA was extracted using mirVana™ miRNA Isolation Kit (Ambion, Ref: AM1560) and then a 2-step qRT–PCR was performed—1^st: cDNA synthesis of each miRNA with stem-loop specific primers using TaqMan™ MicroRNA Reverse Transcription Kit (Applied Biosystems; Ref: 4366596) together with the primer from TaqMan™ MicroRNA Assay (for mature let-7, we used RT-hsa-let-7a; Thermo Fisher, Ref: 4427975); 2^nd: real-time quantification using PCR because of the chain elongation produced by the stem-loop primer unfolding, this allows the binding of two primers and one probe needed for the quantification, and this second step was performed using TaqMan™ MicroRNA Assay (TM-hsa-let-7a; Thermo Fisher, Ref: 4427975) together with the TaqMan™ Gene Expression Master Mix (Applied Biosystems, Ref: 4369016). All reactions were performed in technical duplicates. The amplification efficiency for each primer pair and the $C_t$ were determined automatically by the StepOne Software, v2.2.2 (Applied Biosystems).

## Design and validation of TuDs and siRNAs

In order to achieve the downregulation of miRNAs, we designed specific Tough Decoys for *let-7* as described (Haraguchi *et al*, 2009; Yoon *et al*, 2017). Two stem structures with a single miRNA-binding sites (MBS) linked by linkers were designed. MBS contained a mismatch in the center. The oligonucleotide contained two sequences of the same miRNA. We then inserted three nucleotide linkers between the stem sequence and MBS. A Scramble Tough Decoy plasmid, binding no miRNA, was also prepared.
*let-7* Tough Decoy: 5′-GATCCGACGGCGCTAGGATCATCAAGTGA GGTAGTAGGATC TTTGTATAGTTGAAGTATTCTGGTCACAGAATA CAAGTGAGGTAGTAGGATCTTTGTATAGTTGAAGATGATCCTAGC GCCGTCTTTTTTGGAAA-3′

Scramble Tough Decoy: 5′-GATCCGACGGCGCTAGGATCATCAA CTGGGCGTATAGAC ATCTGTGTTCGTTCCAAGTATTCTGGTCACA GAATACAACTGGGCGTATAGACATCTGTGTTCGTTCCAAGATGAT CCTAGCGCCGTCTTTTTTGGAAA-3′.

This cassette was inserted into the BamHI-HindIII sites of pSilencer2.1-U6 puro vector (Ambion) to generate TuD RNA expression plasmids. Plasmid DNA was purified with a NucleoBond Xtra Midi Kit (Cultech, 22740410.50) and resuspended in 10 mM Tris–HCl (pH 8.0). The efficiency of each TuD to decrease the abundance of mature miRNAs was tested by RT–qPCR.

Validated short interfering RNAs (siRNAs) were obtained from Dharmacon: Irs2 ON TARGETplus SMARTpool (Catalog ID: L-040284-01-0020) and ON TARGETplus Non-targeting Control Pool (Catalog ID: D-001810-10-20) and used following the manufacturer's instructions.

## *In utero* electroporation experiments

Mouse embryos were electroporated *in utero* at E12.5. Briefly, pregnant females were deeply anesthetized with isoflurane and the uterine horns exposed; DNA solution (2 μl) was injected into the lateral ventricle using pulled glass micropipettes, and square electric pulses (30V, 50 ms on–950 ms off, 5 pulses) were applied with an electric stimulator (Cuy21EDIT Bex C., LTD) using round electrodes (CUY650P5, Nepa Gene). Plasmid concentrations were as follows: GFP = 0.7 μg/μl; Irs2 = 1 μg/μl; MISSION *let-7a, let-7b, let-7c* mimic (Sigma; HMI0002, HMI0007, and HMI0009) = 20 nM; TuD-Scr = 1.5 μg/μl; TuD *let-7* = 1.5 μg/μl; siRNA Irs2 = 200 μM; and siRNA Crtl = 200 μM. Combinations of these plasmids were used at the same final individual concentrations.

## hiPSC culture

Human iPSCs obtained from ATCC were cultured at 37°C, 5% CO₂ and ambient oxygen level on Geltrex-coated plates in mTeSR™ Plus medium (STEMCELL Technologies, 05825) with daily medium change. For passaging, iPSC colonies were incubated with ReLSR™ (STEMCELL Technologies, 05872) following the manufacturer's protocol. Pieces of colonies were washed off with DMEM/F12, centrifuged for 5 min at 300 *g* and resuspended in mTeSR1 supplemented with 10 μM Rock inhibitor Y-27632 for the first day.

## Generation of human cerebral organoids

Cerebral organoids were generated as previously described (Lancaster & Knoblich, 2014). Briefly, mycoplasma-free iPSCs were dissociated into single cells using StemPro Accutase Cell Dissociation Reagent (A1110501, Life Technologies) and plated in the concentration of 9,000 single iPSCs/well into low attachment 96-well tissue culture plates in hES medium (DMEM/F12 GlutaMAX supplemented with 20% knockout serum replacement, 3% ES grade FBS, 1% non-essential amino acids, 0.1 mM 2-mercaptoethanol, 4 ng/ml bFGF, and 50 μM Rock inhibitor Y27632) for 6 days in order to form embryoid bodies (EBs). Rock inhibitors Y27632 and bFGF were removed on the 4^th day. On day 6, EBs were transferred into low attachment 24-well plates in NIM medium (DMEM/F12GlutaMAX supplemented with 1:100 N2 supplement, 1% non-essential amino acids, and 5 μg/ml heparin) and cultured for

additional 6 days. On day 12, EBs were embedded in Matrigel drops and then they were transferred in 10-cm tissue culture plates in NDM minus A medium (DMEM/F12 GlutaMAX and Neurobasal in ratio 1:1 supplemented with 1:100 N2 supplement 1:100 B27 without vitamin A, 0.5% non-essential amino acids, insulin 2.5 μg/ml, 1:100 antibiotic–antimycotic, and 50 μM 2-mercaptoethanol) in order to form organoids. 4 days after, Matrigel embedding cerebral organoids were transferred into an orbital shaker and cultured until electroporation in NDM plus A medium (DMEM/F12 GlutaMAX and Neurobasal in ratio 1:1 supplemented with 1:100 N2 supplement 1:100 B27 with vitamin A, 0.5% non-essential amino acids, insulin 2.5 μg/ml, 1:100 antibiotic–antimycotic, and 50 μM 2-mercaptoethanol). During the whole period of cerebral organoid generation, cells were kept at 37°C, 5% CO$_2$ and ambient oxygen level with medium changes every other day. After transferring the cerebral organoids onto the shaker, medium was changed twice per week.

### Electroporation of human cerebral organoids

Cerebral organoids were kept in antibiotics-free conditions prior to electroporation. Electroporations were performed in cerebral organoids at 40 days stages after the initial plating of the cells and fixed 7 days post-electroporation. During the electroporation, cerebral organoids were placed in an electroporation chamber (Harvard Apparatus, Holliston, MA, USA) under a stereoscope and using a glass microcapillary 1–2 μl of plasmid DNAs was injected together with Fast Green (0.1%, Sigma) into different ventricles of the organoids. Plasmid DNA concentrations were as follows: *Gfp* (0.7 μg/μl), *Irs2* (1 μg/μl), TUD-*Scr* (1 μg/μl), and TUD-*let7* (1 μg/μl). Cerebral organoids were subsequently electroporated with five pulses applied at 80V for 50 ms each at intervals of 500 ms (ECM830, Harvard Apparatus). Following electroporation, cerebral organoids were kept for additional 24 h in antibiotics-free media and then changed into the normal media until fixation. Cerebral organoids were fixed using 4% PFA for 20 min at 4°C, cryopreserved with 30% sucrose, and stored at −20°C. For immunofluorescence, 20-μm cryosections were prepared.

### Image analysis, quantification, and statistics

Images were acquired using a florescence microscope (Zeiss Axio Imager Z2) with Apotome-2 and coupled to two different digital cameras (AxioCam MRm and AxioCam ICc) or an inverted confocal microscope (Olympus FluoView FV1000). All images were analyzed with ImageJ (Fiji) or Imaris® software. Telencephalic volume was measured stereologically on serial coronal sections using an upright microscope (Leica DM4000) and Stereo Investigator® software (MBF Bioscience). For three-dimensional visualization of rosettes, brains were inmunostained *in toto* against Pax6, clarified using 3DISCO (Erturk *et al*, 2012), confocal image stacks were obtained, and acquired images were analyzed and processed using Imaris 8 software. For co-localization studies, single-plane images were obtained using a confocal microscope (Olympus FluoView FV1000). For the quantification of fluorescence intensity, images were converted to 8-bit, and intensity was measured with Fiji software using the "Measure/Area integrated intensity and Mean gray value" function. Values were normalized by background intensity. Statistical analysis was carried out in GraphPad Prism 6 Software (GraphPad Software Inc., La Jolla, CA, USA, www.graphpad.com) using ANOVA with post hoc Bonferroni correction (equal variances) or the Welch test with post hoc Games–Howell (different variances), two-way ANOVA, and Tukey's multiple comparisons test (fluorescence intensity analysis), chi-square test, pairwise *t*-test, or independent-samples *t*-test, where appropriate and upon normality testing as indicated. Significance was set at $P = 0.05$. All values represent mean ± standard error of the mean (SEM).

## Data availability

The raw and processed datasets for both RNA-Seq and small RNA-Seq produced in this study are available in the following database:

RNA-Seq data: Gene Expression Omnibus GSE151150 (https://www.ncbi.nlm.nih.gov/geo/query/acc.cgi?acc = GSE151150).

**Expanded View** for this article is available online.

## Acknowledgements

We thank G. Exposito and V. Villar for excellent assistance with imaging, J. Galcerán, E. Herrera, and O. Marín for plasmids, O Marín for the TdTomato reporter mouse colony, R. Hindges for the *Rx3-Cre* and *Dicer*[F/F] colonies, M. Blasco for the *p53*[F/F] colony, and M. Drukker for hiPSCs. We also thank members of our laboratory for insightful discussions and critical reading of the manuscript. V.F. was recipient of an FPI fellowship from the Spanish State Research Agency (AEI), and A.P.-C. was recipient of a predoctoral fellowship from Fundación Tatiana Pérez de Guzmán el Bueno. This work was supported by grants to V.B. from AEI (SAF2015-69168-R, PGC2018-102172-B-I00) and European Research Council (309633). V.B. acknowledges financial support from the AEI, through the "Severo Ochoa" Program for Centers of Excellence in R&D (Ref. SEV-2017-0723).

## Author contributions

VF, MAM-M, and VB conceived and designed the experiments; VF, MAM-M, AP-C, MD, RS, AC, UT, and YN performed and analyzed experiments. J-PL-A, FC, and VB supervised analyses, and provided reagents and resources. VB supervised experiments, provided funding, and wrote the manuscript with input from all other authors.

## Conflict of interest

The authors declare that they have no conflict of interest.

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
