## [Review Process File · The EMBO Journal]

Repression of Irs2 by let-7 miRNAs is essential for homeostasis of the telencephalic neuroepithelium

Virginia Fernández, María-Ángeles Martínez-Martínez, Anna Prieto-Colomina, Adrián Cárdenas, Rafael Soler, Martina Dori, Ugo Tomasello, Yuki Nomura, Jose López-Atalaya, Federico Calegari, and Victor Borrell

DOI: [10.15252/embj.2020105479](https://doi.org/10.15252/embj.2020105479)

Corresponding author(s): Victor Borrell (vborrell@umh.es)

Review Timeline:

Submission Date:	1st May 20
Editorial Decision:	4th Jun 20
Revision Received:	31st Jul 20
Editorial Decision:	17th Aug 20
Revision Received:	21st Aug 20
Accepted:	28th Aug 20

Editor: Karin Dumstrei

Transaction Report:

Dear Victor,

Thank you for submitting your manuscript to The EMBO Journal. Your study has now been seen by two referees and their comments are provided below.

As you can see from the comments, both referees find the study interesting and suitable for publication here. They raise a number of different concerns that would be good to resolve. I am happy to discuss the raised points further and maybe it would be most helpful to do so via phone, email or video.

When preparing your letter of response to the referees' comments, please bear in mind that this will form part of the Review Process File, and will therefore be available online to the community. For more details on our Transparent Editorial Process, please visit our website:

<https://www.embopress.org/page/journal/14602075/authorguide#transparentprocess>

Thank you for the opportunity to consider your work for publication. I look forward to discussing the revisions further.

with best wishes

Karin

Karin Dumstrei, PhD
Senior Editor
The EMBO Journal

- a point-by-point response to the referees' comments, with a detailed description of the changes made (as a word file).
- a word file of the manuscript text.
- individual production quality figure files (one file per figure)
- a complete author checklist, which you can download from our author guidelines

(<https://www.embopress.org/page/journal/14602075/authorguide>).

- Expanded View files (replacing Supplementary Information)

Further information is available in our Guide For Authors:

The revision must be submitted online within 90 days; please click on the link below to submit the revision online before 2nd Sep 2020.

Referee #1:

In this very interesting manuscript the authors describe a novel role for microRNAs at early stages of telencephalon development. Early deletion of Dicer results in a massive phenotype in the rostral cortex with many rosettes & cell death. The authors demonstrate that this is due on the one hand to aberrant p53 signalling as the phenotype is completely rescued by p53 deletion, and on the other hand up-regulation of *Irs2* via loss of let-7 mediated inhibition is crucial. The authors show that overexpression of *Irs2* in murine and human cells recapitulates the over-proliferation phenotype and co-electroporation with let7 rescues this. Finally, they demonstrate that reducing let7 also phenocopies the rosette formation. These are very novel and important data highlighting for the first time a key role of specific microRNAs in regulating, i.e. limiting, NSC amplification at early stages. Moreover, the molecular cascade unravelled by the authors is reminiscent of some early childhood cancers. However, a few important sets of data are missing to fully substantiate the working hypothesis and to understand the regionalization of the phenotype.

Major suggestions:

- 1) The model is that *Irs2* is up-regulated by stress and down-regulated by let7. In the p53 dicer double mutants, the former is improved, but the latter is still missing. This prompts the question if *Irs2* levels are lower in these double mutants. Ideally the authors could stain for *Irs2* in double-mutants that are virtually completely rescued (40%, p.15) and those that still have some cell death. In regard to this incomplete penetrance it may be difficult to see how much transcriptional changes are normalized in the double KO despite the lack of all these microRNAs. But if doable it would be really interesting.
- 2) The other major question remaining is the regionalization of the phenotype - in the experiments with let7 Tough Decoys and/or the *Irs2* overexpression, what happens if electroporations were not rostral? Any phenotype?

Minor suggestions:

- a) I do not understand why the authors start with the minor (not to say a bit boring) and late phenotype in the cortex, and only in Figure 2-3 arrive at the interesting, novel and early phenotype. I would favour to start with an early phenotype as this is normally causative and anything later possibly indirect. In addition, the cortex phenotype has been studied by various other Cre-lines - so this is something for discussion and hence Figure 1 can be moved entirely to supplemental material (in my opinion). Moreover, one notices in Figure 1A that the OB appears to be missing, but this is only followed up in Figure 3. Please rearrange.
- b) Figure 6M the section is out of focus/blurry.
- c) For data not normally distributed median and IQR may be the better representation.
- d) Please discuss how the adhesion belt may be affected, as not many mRNAs nor targets of reduced miRNAs related to adhesion were shown or detected. One interpretation could be that only the overproliferation causes rosette formation without specifically altering the adhesion belt - just as a consequence of too many mitosis?

Referee #2:

Fernandez et al. explore the role of miRNAs in the early forebrain development by using conditional knockout of Dicer at an earlier stage than in previous studies. In addition to confirming increased apoptosis shown by others, this approach revealed a striking new phenotype with abundant hyperproliferative rosettes in different areas of the telencephalon. The authors provide a detailed characterization of this phenotype and use a series of genetic manipulations to further dissect the downstream *let-7* / *p53* / *lsl2* pathway mediating rosette formation. The study is well-designed, with convincing data, very detailed, well presented, and has interesting translational implications for pediatric brain tumors.

Major comments:

1. The present findings are in contrast to previous reports by De Pietri Tonelli et al., *Development* 2008, Kawase-Koga et al., *Dev Dyn* 2009, and Saurat et al., *Neural Dev* 2013, which, in addition to cortical thinning, described a disruption of cortical layering due to degeneration of upper cortical layer cells and/or overproduction of deep layer cells. In addition, previous studies observed massive cell death in the neocortex that extended to later developmental stages. The absence of these phenotypes in the current study is surprising, as gene ablation was performed earlier than in the previous ones. The authors do acknowledge some discrepancies with earlier publications, but a clear explanation for them is missing. As the differences likely have to do with incomplete gene ablation by Rx-Cre in the dorsal telencephalon, it would be helpful to quantify the efficiency of Rx-Cre recombination (assessed qualitatively by *in situ* in Fig. S2) in different regions of the telencephalon, and to clearly acknowledge the specific properties of the Cre line used.
2. Findings illustrated by representative images in Fig. 6A-F (phospho-p53 and caspase-3 stainings) should also be supported by quantitative analyses.
3. The authors conclude that rosette formation upon *lsl2* overexpression is not linked to apoptosis. However, at the age when these experiments were done, E13.5, the wave of massive apoptosis linked to loss of miRNAs is expected to be almost over, according to the data presented in Fig. 2E. In addition, an increase in apoptotic cells, albeit mild, can indeed be seen in Fig. 7L. To reach the

proposed conclusion that *lsr2*-dependent rosette formation occurs in the absence of apoptosis, the experiment should be performed at an earlier stage and the results should be quantified.

4. The authors demonstrate that gain-of-function of *lsr2* is sufficient to induce rosettes (Fig. 7). However, it remains unclear whether the formation of rosettes in the *Dicer* mutant and in *TuD let-7* embryos is mediated by *lsr2*. For this, *lsr2* loss-of-function experiments would be required. In addition, it would be helpful to demonstrate the upregulation of *lsr2* in *TuD let-7* embryos and organoids. Although known from the literature, it has not been shown in the context of the systems used in this study.

5. The proposed relationship between the p53 pathway and *lsr2* upregulation is not explored in the paper. The statements about *lsr2* expression being boosted by the overactivated p53 pathway should therefore be toned down.

Minor remark:

1. Fig. 4B - figure legend should mention what the arrows indicate.

Response to reviews on manuscript EMBO J-2020-105479

We are very thankful to both reviewers for very kindly acknowledging the high scientific interest, timeliness, novelty and importance of our results, as well as their interest and relevance in early childhood cancers. Reviewer #2 even further states that **“The study is well-designed, with convincing data, very detailed, well presented, and has interesting translational implications for pediatric brain tumors.”** We really appreciate these positive comments.

Following is a point-by-point response to their individual major and minor concerns:

Referee #1:**Reviewer's comment:**

1) The model is that *Irs2* is up-regulated by stress and down-regulated by *let7*. In the p53 dicer double mutants, the former is improved, but the latter is still missing. This prompts the question if *Irs2* levels are lower in these double mutants. Ideally the authors could stain for *Irs2* in double-mutants that are virtually completely rescued (40%, p.15) and those that still have some cell death. In regard to this incomplete penetrance it may be difficult to see how much transcriptional changes are normalized in the double KO despite the lack of all these microRNAs. But if doable it would be really interesting.

Author's response:

We thank this reviewer for highlighting this important point, and we completely agree on the high interest of obtaining these results with regard to our study. Unfortunately, due to the strong impact of Covid-19 in our country and the corresponding safety measures taken at the animal facility of our research Institute (maximal reduction of mouse colonies to allow minimal personnel activity), we did not have access to double mutant embryos within the timeframe of this revision. Thus, we have been completely unable to respond to this point experimentally. Nevertheless, we have obtained new results showing that in Rx-Dicer single mutant embryos there is remaining expression of *let-7* at the embryonic stage of rosette formation. This is very important because there is published evidence showing that p53 negatively regulates *let-7* expression and binding to Argonaute (Saleh et al., 2011; Hau et al., 2012; Krell et al., 2016; Subramanian et al., 2015). Hence, in double mutants, the loss of p53 may allow partially higher levels of *let-7*, and this combined with the already remaining *let-7* expression prior to p53 loss, is likely sufficient to limit *Irs2* levels and prevent rosette formation in these double mutants.

Action taken – We have added new Results showing that in Rx-Dicer single mutant embryos there is remaining (although significantly lower) expression of *let-7*, presented in the Results section (page 13, lines 377-378) and in the new Figure 4J of the revised manuscript. In light of published evidence and these new results, in Discussion of the revised manuscript we now present the above argument (page 23, lines 680-690).

2) The other major question remaining is the regionalization of the phenotype - in the experiments with *let7* Tough Decoys and/or the *Irs2* overexpression, what happens if electroporations were not rostral? Any phenotype?

We thank this reviewer for bringing up this important point. Indeed, in the previous version of our manuscript we reported that in *Rx-Dicer* mutants rosettes systematically formed in the most rostral part of the dorsal telencephalon, with the rest of the cerebral cortex remaining essentially unaffected. This is now better documented in the new Figure EV3. This regionalization of the phenotype could result from the fact that *Rx*-driven Cre recombination is not homogeneous across the telencephalon (see new Figure EV1, and point 1 from reviewer #2), or because the rest of the cerebral cortex is unaffected by the loss of *let-7* / increased *Irs2*, as pointed here by this reviewer. To address this question, we have overexpressed *Irs2* by *in utero* electroporation affecting more than half of the rostro-caudal extent of the cerebral cortex. This experiment shows that the phenotype of neuroepithelial disorganization and formation of rosettes is essentially restricted to the most rostral part of the cortex (internal control), while the adjacent and parietal cerebral cortex remain unaffected.

Action taken – We have added a new extended view figure (Figure EV3) showing, on the one hand, examples of *Rx-Dicer* mutants at E12.5 and E17.5 with neuroepithelial disorganization and rosettes only occurring in the most rostral aspect of the dorsal telencephalon. In addition, Figure EV3 also shows an example of a very extensive electroporation demonstrating that increased expression of *Irs2* leads to the formation of rosettes in a strictly regionalized manner, only in the most rostral but not the further caudal cortical primordium. These new results are also described in a new, dedicated paragraph in the relevant Results section (page 16, lines 456-470).

Minor suggestions:

a) I do not understand why the authors start with the minor (not to say a bit boring) and late phenotype in the cortex, and only in Figure 2-3 arrive at the interesting, novel and early phenotype. I would favour to start with an early phenotype as this is normally causative and anything later possibly indirect. In addition, the cortex phenotype has been studied by various other Cre-lines - so this is something for discussion and hence Figure 1 can be moved entirely to supplemental material (in my opinion). Moreover, one notices in Figure 1A that the OB appears to be missing, but this is only followed up in Figure 3. Please rearrange.

We sincerely thank this reviewer for this thoughtful and constructive comment, and indeed, we agree that the proposed order of presentation is likely more exciting to the reader and more temporally logical. Reviewer #2 also comments on the cortical phenotype, and also highlights that this should be discussed more explicitly as our report on *Rx-Dicer* mutants is different from defects reported previously using other Cre lines. In the revised manuscript, we have changed the order in which results are presented as suggested by reviewer #1. Given the new order of data presentation in the manuscript, the apparent absence of OB is now mentioned first together with the rest of data in the new Figure 2 (previous Figure 3), and the description of the late phenotype in the cortex shown in the previous Figure 1 (now Figure

EV4) is presented last, in Discussion. Hence, the suggested order rearrangement with respect to the OB phenotype is no longer necessary. We also discuss in some detail the fact that this embryonic cortical phenotype in *Rx-Dicer* mutants differs from that previously reported when using other Cre lines, as requested by reviewer #2.

Action taken – In the revised version of the manuscript, the original Figure 1 has become Figure EV5, and the results there presented and their relevance as compared to previous studies are now briefly described in a new paragraph in Discussion (pages 21, 22; lines 598-633; see also comment 1 by reviewer #2).

b) Figure 6M the section is out of focus/blurry.

Action taken – The previous image has been replaced by a focused version, as suggested. In the revised manuscript, this panel now corresponds to Figure 5O.

c) For data not normally distributed median and IQR may be the better representation.

We thank the reviewer for his/her concern regarding the most correct representation of our data. Unfortunately, for small sample sizes normality tests have little power to reject the null hypothesis and therefore it is unlikely that they would detect non-normality, hence not being accurate. However, we are also well aware that many different data distributions can lead to the same bar or line plots. To overcome these limitations and to allow readers to critically evaluate our results, we show all individual data points in every graph, in addition to the mean and standard error of each dataset.

Action taken – All datasets in the revised manuscript have been represented with their individual data points, in addition to the mean and standard error.

d) Please discuss how the adhesion belt may be affected, as not many mRNAs nor targets of reduced miRNAs related to adhesion were shown or detected. One interpretation could be that only the overproliferation causes rosette formation without specifically altering the adhesion belt - just as a consequence of too many mitosis?

Again, we are very thankful to this reviewer for bringing to our attention this weakness in our manuscript. Unfortunately, we failed to properly highlight that indeed there are a number of both differentially-expressed mRNAs and targets of differentially-expressed miRNAs that are functionally related to cell adhesion, and thus possibly contribute to altering the adhesion belt. Some of these results were shown in Figure 5 of the original manuscript (current Figure 4), and presented in Supplementary Tables, which due to the large amount of different tables and data contained may have gone unnoticed to this reviewer.

Briefly, functional annotation of targets of differentially expressed miRNAs revealed many categories related to adhesion including the following GO categories:

“Heterophilic cell-cell adhesion via plasma membrane cell adhesion molecules” (GO:0007157; p value= 0.0044; genes: PTPRD, CADM2, TENM2, ITGA4, CD164)
 “Substrate adhesion-dependent cell spreading” (GO:0034446; p value= 0.093; genes: SRCIN1, ITGA4, EPHB3)

We also found significant enrichment for the KEGG pathway

“Focal adhesion” (mmu04510; p value = 8.58E-05; genes: AKT1, IGF1R, COL4A2, CCND1, VAV3, BRAF, TLN2, ROCK2, PDGFRB, ITGA4, THBS1, FLNA).

Many differentially expressed mRNAs in Rx-Dicer mutant embryos were functionally related to:

“Cell adhesion” (GO:0007155; p value = 0.0021; genes: COL18A1, B4GALT1, PTPRK, RET, CADM4, PTPRM, LPP, PODXL, EFNB2, ITGA2, CD164, MYH9, CDH4, KITL, IGSF11, TENM2, CX3CR1, TTYH1, EMB, NCAN, CHL1, THBS4, SPON1)
 “Heterophilic cell-cell adhesion via plasma membrane cell adhesion molecules” (GO: 0007157; p value = 0.0319; genes: CADM4, TENM2, CD164, CDH4, LGALS9)
 “Cell-cell adhesion” (GO:0098609; p value = 0.0320; genes: TMEM47, TACSTD2, PDLIM5, BAG3, PPL, WASF2, DOCK9, GIPC1, GPRC5A, EHD4)
 “Single organismal cell-cell adhesion” (GO:0016337; p value = 0.0394; genes: TENM2, ADGRV1, TTYH1, EFR3A, MYH9, SOX9, CD2AP)
 “Extracellular matrix-receptor interaction” (mmu04512; p value = 0.0539; genes: ITGB8, HSPG2, ITGA2, COL2A1, COL4A6, THBS4)

Gene set enrichment analysis (GSEA) of differentially expressed mRNAs revealed significant enrichment for the GO categories:

“Biological adhesion” (FDR q-val = 0.011; ABI3BP, ADGRV1, B4GALT1, BCL2L11, CADM4, CCN2, CD2AP, CD164, CDH4, CDH19, CDK6, CHL1, COL18A1, CSF1, CX3CR1, EFNB1, EFNB2, EFR3A, EMB, FBLN1, FNDC3B, IGSF11, IRF1, ITGA2, ITGB8, KITL, LGALS9, LPP, MYH9, MYO1F, NCAN, ONECUT1, ONECUT2, PALLD, PCDH9, PODXL, PRTG, PTEN, PTPRJ, PTPRK, PTPRM, RET, RPL29, SHH, SOX9, SPON1, STX3, TACSTD2, TENM2, THBS4, TMEM47, TTYH1, VAV1, ZBTB16, ZFHX3)
 “Positive regulation of cell adhesion” (FDR q-val = 0.013; IRF1, RET, EFNB1, EFNB2, ZBTB16, CSF1, SHH, STX3, PODXL, ABI3BP, VAV1, ZFHX3, ITGA2, PTPRJ, FBLN1, CDK6, LGALS9)
 “Regulation of cell adhesion” (FDR q-val = 0.032; IRF1, RET, EFNB1, EFNB2, ZBTB16, CSF1, SHH, STX3, PODXL, ABI3BP, VAV1, ZFHX3, ITGA2, PTPRJ, FBLN1, CDK6, LGALS9, ONECUT1, ONECUT2, SOX9, PTEN, TACSTD2, MYO1F)
 “Cell-cell adhesion” (FDR q-val = 0.069; PRTG, CD2AP, IRF1, RET, EMB, EFR3A, PALLD, EFNB1, CD164, MYH9, EFNB2, ZBTB16, ADGRV1, TENM2, LPP, SHH, CDH19, PTPRM, CX3CR1, SOX9, CDH4, TMEM47, THBS4, PODXL, TTYH1, CHL1, IGSF11, LGALS9)

GSEA for KEGG pathways identified genes associated to:

“Focal adhesion” (COL2A1, COL4A6, MYLK, ITGB8, PIK3R3, PTEN, THBS4, VAV1, ITGA2)
 “Tight junction” (HCLS1, MAP3K20, MYH9, PTEN)
 “Adherens junction” (PTPRM, WASF2, PTPRJ)

Action taken – A synthesis of the above results has been added to the Results section in the revised manuscript (page 13, lines 360-375), supported by two new Supplementary Tables with the GSEA-derived categories (Tables S9 and S10), and a new supplementary table specifically containing the above detailed information related to cell adhesion (Table S11).

Referee #2:

Major comments:

1. The present findings are in contrast to previous reports by De Pietri Tonelli et al., Development 2008, Kawase-Koga et al., Dev Dyn 2009, and Saurat et al., Neural Dev 2013, which, in addition to cortical thinning, described a disruption of cortical layering due to degeneration of upper cortical layer cells and/or overproduction of deep layer cells. In addition, previous studies observed massive cell death in the neocortex that extended to later developmental stages. The absence of these phenotypes in the current study is surprising, as gene ablation was performed earlier than in the previous ones. The authors do acknowledge some discrepancies with earlier publications, but a clear explanation for them is missing. As the differences likely have to do with incomplete gene ablation by Rx-Cre in the dorsal telencephalon, it would be helpful to quantify the efficiency of Rx-Cre recombination (assessed qualitatively by in situ in Fig. S2) in different regions of the telencephalon, and to clearly acknowledge the specific properties of the Cre line used.

We agree that a more detailed and careful characterization of the Rx-Cre mouse line is important to put our new findings in context, especially in relation to differences with previous studies of cortical development in Dicer mutants using other Cre-driver mouse lines.

Actions taken – We have used a TdTomato reporter mouse line to quantify the efficiency of Rx-Cre recombination in the different regions of the developing telencephalon most relevant to our study. These new quantifications are presented in Figure EV1 and new Table S1, and described in Results to clearly detail the specific properties of the Cre mouse line used in this study: page 6, lines 138-151.

In light of these new more detailed analyses, we have added a sentence in Discussion providing a potential explanation for discrepancies between our results and phenotypes reported by using later-recombining Cre lines, in line with the argument presented here by this reviewer (page 21, lines 629-633).

2. Findings illustrated by representative images in Fig. 6A-F (phospho-p53 and caspase-3 stainings) should also be supported by quantitative analyses.

We originally considered that the results presented in those images were sufficiently compelling to not require a quantitative assessment. However, we agree that a quantitative analysis goes quite beyond simply illustrating a result, and thus we have followed this

reviewer's advice and performed these quantifications. We thank this reviewer for raising this point, as we agree that the new results strengthen the value of our previous qualitative observations.

Action taken – We have quantified phospho-p53⁺ and caspase-3⁺ cells as suggested, and included the results in Figure 5, panels C and H, respectively.

3. The authors conclude that rosette formation upon *Irs2* overexpression is not linked to apoptosis. However, at the age when these experiments were done, E13.5, the wave of massive apoptosis linked to loss of miRNAs is expected to be almost over, according to the data presented in Fig. 2E. In addition, an increase in apoptotic cells, albeit mild, can indeed be seen in Fig. 7L. To reach the proposed conclusion that *Irs2*-dependent rosette formation occurs in the absence of apoptosis, the experiment should be performed at an earlier stage and the results should be quantified.

We are afraid that this reviewer was a bit confused with the nature of this set of experiments and their interpretation. To recapitulate, in our study we first show that the rosette phenotype in *Rx-Dicer* mutant embryos is linked to massive apoptosis and increased proliferation. Our transcriptomic analyses in *Rx-Dicer* mutant embryos reveal that this correlates with expression changes in many genes, including increased *Irs2* and p53 signaling, the former explaining proliferation and the latter explaining massive apoptosis. To test if high levels of *Irs2* alone is sufficient to induce formation of rosettes, we decided to move away from *Rx-Dicer* mutant embryos. We performed *in utero* electroporation to overexpress *Irs2* in wild-type embryos at E12.5, where all other parameters along development are otherwise normal, including minimal apoptosis at any previous stage. Analysis at E14.5 of these electroporations showed that *Irs2* overexpression was indeed sufficient to induce rosettes. Then, to rule out that *Irs2* overexpression might itself induce massive apoptosis, and this be the mechanism of rosette formation, we repeated *Irs2* overexpression in wild-type embryos and analyzed only one day after onset of overexpression, at E13.5, while rosettes are forming. These results were presented in the former Figure 7I-L (new Figure 6 in the revised manuscript) and show very low levels of apoptosis, much lower than in *Rx-Dicer* mutants, which is virtually absent among progenitor cells. Hence, we conclude that rosette formation upon *Irs2* overexpression is not linked to apoptosis.

Nevertheless, we agree with this reviewer that these results should be quantified, and we have measured the abundance of Caspase3⁺ apoptotic cells. This quantification has been performed separately in VZ and CP to distinguish apoptosis in progenitor cells (VZ), as in *Rx-Dicer* mutants, or in neurons (CP). Our results confirm the virtual absence of apoptosis in VZ, at a level similar to control-electroporated embryos, and a significant but very low increase in apoptosis in CP. This is very different from *Rx-Dicer* mutants, where apoptosis affects mostly progenitor cells and the rate of cell death is one order of magnitude higher: average of 4 casp3⁺ cells /100mm in *Irs2* overexpression, compared to 40-60 casp3⁺ cells in *Rx-Dicer* mutants at the onset of rosette formation (see new Figure 1).

Action taken – We have added a new panel M to Figure 6 (former Figure 7) reporting the quantification of Caspase-3⁺ cell abundance in CP and VZ of E13.5 wild-type embryos overexpressing GFP or *Irs2* from E12.5. To avoid confusion on this point among the

readership, we have also made more clear in Results that these experiments were performed in wild-type embryos, and thus demonstrate that cell death is not involved in the rosette phenotype (page 16, line 446; page 17, lines 481, 487 and 489-494).

4. The authors demonstrate that gain-of-function of *Irs2* is sufficient to induce rosettes (Fig. 7). However, it remains unclear whether the formation of rosettes in the *Dicer* mutant and in *TuD let-7* embryos is mediated by *Irs2*. For this, *Irs2* loss-of-function experiments would be required.

We completely agree with this reviewer that rescue of the *Rx-Dicer* mutant phenotype is of course a very interesting and exciting experiment. Unfortunately, we attempted this in the past by *in utero* electroporation at E11.5 and E12.5, but *Rx-Dicer* mutants never survived the aggressive *in utero* electroporation procedure, likely because they are extremely fragile (frequently die already at E13.5 without experimental manipulation). However, we have performed the alternative experiment proposed by this reviewer, co-electroporating wild-type embryos with *TuD let-7* and siRNAs against *Irs2*, for loss-of-function. The results of these new experiments show a complete rescue of the rosette phenotype in a majority of *TuD let-7* embryos, demonstrating the primary genetic mechanism of formation of rosettes in *Rx-Dicer* mutants: increase of *Irs2* caused by a loss of *let-7*.

Action taken – We have added a new Figure 7 showing the effects of expressing *TUD let-7*, and how this rosette phenotype is rescued by *Irs2* loss-of-function. These new findings are now described in the Results section (page 18, lines 525-532) and discussed in pages 23,24 (lines 676-690).

In addition, it would be helpful to demonstrate the upregulation of *Irs2* in *TuD let-7* embryos and organoids. Although known from the literature, it has not been shown in the context of the systems used in this study.

The reviewer is absolutely right that upregulation of *Irs2* upon loss of *let-7* has never been demonstrated in the context of embryonic development of the telencephalon, which is yet one more point of novelty of our study. To demonstrate this, we have performed immunostains against *Irs2* in mouse embryos electroporated with *TuD-let-7*, and quantified the fluorescence intensity of *Irs2* antibody stain. These results show the significant upregulation of *Irs2* protein in the electroporated area compared to the adjacent, non-electroporated area. This reviewer also requests that we perform the same type of *Irs2* abundance analysis in sections from our *TuD-let-7* human cerebral organoids, *in vitro*. However, in our modest opinion this becomes less important once we already demonstrate this point in the telencephalon of living embryos, *in vivo*, especially since we already show in human cerebral organoids that gain of *Irs2* and loss of *let-7* produce the same effect, as in mouse. Hence, we have taken no action on this second point.

Action taken – We have added a new supplementary Figure EV4 showing the increased abundance of *Irs2* by immunofluorescence upon expression of *TuD-let-7*, and this new finding is presented in the Results section (page 18, lines 523-525).

5. The proposed relationship between the p53 pathway and Irs2 upregulation is not explored in the paper. The statements about Irs2 expression being boosted by the overactivated p53 pathway should therefore be toned down.

Action taken – As suggested by this reviewer, we have toned down our statements on the boosting of Irs2 expression as a result of overactivated p53 signaling, in page 14 (line 384), page 17 (lines 474, 475) and page 19 (line 552).

Minor remark:

1. Fig. 4B - figure legend should mention what the arrows indicate.

We thank the reviewer for pointing at this omission.

Action taken – The mistake has been fixed in the revised manuscript, where former Figure 4 corresponds to the new Figure 3 (page 42).

Dear Victor,

Thank you for submitting your revised manuscript to The EMBO Journal. Your study has now been seen by referee #1. As you can see below, the referee appreciates the introduced changes and support publication here.

I am therefore very pleased to let you know that we will accept the manuscript for publication here. Before sending you the formal acceptance letter there are just a few editorial issues that needs to be resolved in a final revision.

- The EV tables should each have a legend - please add as separate tab in the excel files
- For the reference list please shorten to 10 authors et al.
- We can only have 3-5 keywords
- It looks like that Fig 3A is reused in 5M-N please indicate this in the figure legends as well.
- Please double check that there are scale bars
- Make sure to remove the password for the GEO submitted data set
- We include a synopsis of the paper (see <http://emboj.embopress.org/>). Please provide me with a general summary statement and 3-5 bullet points that capture the key findings of the paper.
- We also need a summary figure for the synopsis. The size should be 550 wide by [200-400] high (pixels) also OK to use something from the figures
- I have asked our publisher to do their pre-publication checks on the paper. They will send me the file within the next few days. Please wait to upload the revised version until you have received their comments.

That should be all - you can use the link below to upload the final version.

Congratulations on a nice study

With best wishes

Karin

Karin Dumstrei, PhD
Senior Editor
The EMBO Journal

- a point-by-point response to the referees' comments, with a detailed description of the changes made (as a word file).

- a word file of the manuscript text.

- individual production quality figure files (one file per figure)

- a complete author checklist, which you can download from our author guidelines

(<https://www.embopress.org/page/journal/14602075/authorguide>).

- Expanded View files (replacing Supplementary Information)

Further information is available in our Guide For Authors:

The revision must be submitted online within 90 days; please click on the link below to submit the revision online before 15th Nov 2020.

Referee #1:

The authors have addressed all my suggestions, most importantly adding new data on the clear region-specific effect of dicer deletion and *Irs2* electroporation restricted to rostral telencephalon. This is very interesting and further strengthens the impact (and novelty) of this work. Also the flow of the manuscript is much improved, such that it will be of great interest for the readers of The EMBO Journal.

Karin Dumstrei
Editor, *The EMBO Journal*

Re: Submission of revised manuscript EMBOJ-2020-105479R

“Repression of Irs2 by let-7 miRNAs is essential for homeostasis of the telencephalic neuroepithelium” by Fernández et al. to *EMBO J*

August 21st, 2020

Dear Karin,

Thanks for your decision letter of August 17th 2020 accepting in principle our manuscript entitled *“Repression of Irs2 by let-7 miRNAs is essential for homeostasis of the telencephalic neuroepithelium”*, by Fernández et al.

As requested, we have resolved the editorial issues raised by your team:

- Each of the EV tables now have a legend (added as separate tab in the excel files).
- The author list has been reduced to 10 authors et al. in References.
- We have reduced our keywords to 5.
- We have indicated in the legend to Figure 5M-N the re-use of images from Fig 3A.
- We have double checked that all scale bars are now indicated.
- We have removed the password for the GEO submitted data set.
- We include a synopsis of the paper with a general summary statement and 4 bullet points that capture the key findings of the paper.
- We include a summary figure for the synopsis, sized within the requested limits.
- We have addressed all the comments from your publisher in the pre-publication checks on the paper. On this regard, I want to explicitly mention that a major comment was that in Figure 1 there is a quantification in panel G for which there is no example in panel F. This is fine and correct, as not every single quantitative analysis in a study is paralleled with example images. Similar cases are found in most (all) papers, including our own in Figure 2H, Figure 3C,D, Figure EV5F-L. This is standard practice completely acceptable in our field.

Looking forward to move on with the publication of this study, with my best regards,

Victor Borrell

Dear Victor,

Thanks for submitting your revised manuscript to The EMBO Journal. I have now had a chance to take a look at everything and I appreciate the introduced changes.

I am therefore very pleased to accept your manuscript for publication here.

Congratulations on a great paper

With best wishes

Karin

Karin Dumstrei, PhD
Senior Editor
The EMBO Journal

Please note that it is EMBO Journal policy for the transcript of the editorial process (containing referee reports and your response letter) to be published as an online supplement to each paper. If you do NOT want this, you will need to inform the Editorial Office via email immediately. More information is available here: http://emboj.embopress.org/about#Transparent_Process

Your manuscript will be processed for publication in the journal by EMBO Press. Manuscripts in the PDF and electronic editions of The EMBO Journal will be copy edited, and you will be provided with page proofs prior to publication. Please note that supplementary information is not included in the proofs.

Should you be planning a Press Release on your article, please get in contact with embojournal@wiley.com as early as possible, in order to coordinate publication and release dates.

If you have any questions, please do not hesitate to call or email the Editorial Office. Thank you for your contribution to The EMBO Journal.

** Click here to be directed to your login page: <http://emboj.msubmit.net>

Corresponding Author Name: Victor Borrell

Journal Submitted to: The EMBO Journal

Manuscript Number: EMBOJ-2020-105479